# Hepatocyte mitochondria-derived danger signals directly activate hepatic stellate cells and drive progression of liver fibrosis

Ping An [1,2,6], Lin-Lin Wei[1,3,6], Shuangshuang Zhao[1,4,5], Deanna Y. Sverdlov[1], Kahini A. Vaid[1], Makoto Miyamoto[1], Kaori Kuramitsu[1], Michelle Lai[1] & Yury V. Popov [1✉]

Due to their bacterial ancestry, many components of mitochondria share structural similarities with bacteria. Release of molecular danger signals from injured cell mitochondria (mitochondria-derived damage-associated molecular patterns, mito-DAMPs) triggers a potent inflammatory response, but their role in fibrosis is unknown. Using liver fibrosis resistant/susceptible mouse strain system, we demonstrate that mito-DAMPs released from injured hepatocyte mitochondria (with mtDNA as major active component) directly activate hepatic stellate cells, the fibrogenic cell in the liver, and drive liver scarring. The release of mito-DAMPs is controlled by efferocytosis of dying hepatocytes by phagocytic resident liver macrophages and infiltrating Gr-1(+) myeloid cells. Circulating mito-DAMPs are markedly increased in human patients with non-alcoholic steatohepatitis (NASH) and significant liver fibrosis. Our study identifies specific pathway driving liver fibrosis, with important diagnostic and therapeutic implications. Targeting mito-DAMP release from hepatocytes and/or modulating the phagocytic function of macrophages represents a promising antifibrotic strategy.

[1] Division of Gastroenterology, Hepatology and Nutrition, Beth Israel Deaconess Medical Center, Harvard Medical School, 330 Brookline Avenue, Boston, MA 02215, USA. [2] Division of Gastroenterology and Hepatology, Renmin Hospital, Wuhan University, 238 Jiefang Road, Wuhan 430060 Hubei, China. [3] Beijing YouAn Hospital, Capital Medical University, No. 8, Xitoutiao, Youanmenwai, Fengtai District, Beijing 100069, China. [4] The Joint Program in Infection and Immunity, Guangzhou Women and Children's Medical Center, Guangzhou 510623, China. [5] Institute Pasteur of Shanghai, Chinese Academy of Science, 320 Yueyang Road, Shanghai 200031, China. [6]These authors contributed equally: Ping An, Lin-Lin Wei. ✉email: ypopov@bidmc.harvard.edu

It is well known that individual patients with virtually any chronic liver disease demonstrate significant differences in the rate of fibrosis progression, ranging widely from 10 to 50 years until cirrhosis develops[1]. A similar phenomenon, the precise mechanism of which is poorly understood, has been reported in inbred mouse strains, where different genetic makeup determines susceptibility to liver fibrosis. This ranges from fibrosis resistance in A/J mice to high susceptibility in the BALB/c strain[2,3]. Understanding the cellular and molecular basis for susceptibility versus resistance to fibrosis/cirrhosis is critical to development of effective antifibrotic therapies, identification of patients who are likely to progress and most in need of antifibrotic treatment, and for stratification of patients in clinical trials[4].

Prior studies provide evidence that a complex interplay of genetic factors, and possibly epigenetic and environmental factors, might determine the wide variation in susceptibility to fibrosis among individuals. Using quantitative trait loci analysis, Hillebrandt et al.[2] identified complement 5 as one of the susceptibility genes in inbred mice, and confirmed that a polymorphism in this gene correlates with fibrosis in human patients with chronic hepatitis C[5]. Another GWAS study in hepatitis C patients identified polymorphisms in the *TLR4* (Toll-like receptor 4) gene as a determinant of fibrosis progression[6]. However, the precise cellular and molecular basis of varying individual susceptibility to fibrosis, as found in all chronic disease irrespective of etiology, remains poorly understood.

Regardless of underlying etiology, the initiating event eventually leading to tissue fibrosis is cell injury and/or death. Sterile cell death and injury (e.g., hepatocytes in the liver) may lead to the release of intracellular molecules called damage-associated molecular patterns (DAMPs). These molecules are recognized by the innate immune system by pattern recognition receptors, often the same molecular sensors that detect pathogens. DAMPs are derived from different subcellular compartments, including mitochondria, which evolved from proteobacteria (engulfed several billions years ago by a eukaryotic cell and adapted as intracellular "endosymbionts")[7]. Due to bacterial origin, many structural components of mitochondria (including its DNA) share significant similarities with bacteria and are considered a major source of highly immunogenic "mito-DAMPs" exposed by injured or dying cells[8]. The liver is extremely rich in mitochondria due to its critical metabolic function in the body, with each hepatocyte containing 1000–2000 mitochondria[9]. Particularly in the liver, mito-DAMPs released from injured hepatocytes may represent one of the most abundant and potent "danger signals" that trigger or perpetuate the innate immune response. However, whether hepatocyte-derived mito-DAMPs impact fibrogenesis in the liver disease is not known.

The complexity of regulatory control of scarring is further increased by the fact that multiple hepatic cell lineages participate in the fibrotic response. While activated hepatic stellate cells/myofibroblasts (HSCs/MFs) are the ultimate fibrogenic effector cell directly responsible for laying down fibrillar collagens[10], the critical paracrine contribution of macrophages[11], hepatic progenitor cells/reactive cholangiocytes[12], and endothelial cells[13] in determining the pace of fibrosis progression is now increasingly recognized. Macrophages in particular have a central but complex role in regulating liver fibrosis[14], exerting opposing roles during progressive and regression stages of liver fibrosis[15]. The mechanistic role of their phagocytic function of efferocytosis (the process of engulfment and removal of dead/dying cells by neighboring phagocytes), the main mechanism that limits inflammatory response to the "danger signals" emanating from damaged or dead cells, in regulating fibrotic outcomes remains obscure.

In this study, we performed in-depth studies of recovery from sub-lethal acute livery injury using an inbred-resistant-/-susceptible mouse strain model system in order to characterize and interrogate the pathophysiologic mechanism responsible for susceptibility to tissue fibrosis. We show that efficient efferocytosis (phagocytosis of injured/dead hepatocytes) by resident F4/80(+) liver macrophages and infiltrating Gr-1(+) myeloid cells prevents the release of hepatocyte-derived mito-DAMPs and is a critical determinant of resistance to hepatic fibrosis. Conversely, prolonged exposure to mito-DAMPs post injury due to inefficient efferocytosis of dead hepatocytes, or exogenous mito-DAMPs administration, is sufficient to trigger fibrogenic activation of HSCs in vivo and in vitro. Mitochondrial DNA (mtDNA), a major active component of mito-DAMPs, is elevated in the sera of non-alcoholic steatohepatitis (NASH) patients and particularly those with significant fibrosis. These findings represent the discovery of a novel regulatory pathway in hepatic fibrosis with important diagnostic and therapeutic implications.

## Results

**Fibrosis-resistant-/-susceptible inbred mouse strain model.** First, we characterized fibrosis susceptibility in three common inbred mouse strains (FVB, C57Bl/6, and BALB/c) subjected to chronic administration of hepatotoxin thioacetamide (TAA) for 6 weeks. FVB demonstrated resistance; BALB/c strain developed the most significant fibrosis; and C57Bl/6 mice showed intermediate susceptibility to liver fibrosis (Supplementary Fig. 1).

Chronic hepatotoxin-induced fibrosis models are characterized by repetitive liver injury, which leads to gradual fibrosis progression over an extended period of time. We hypothesized that such dramatically divergent strain phenotypes might be determined by strain-specific differences in response to each acute insult. To test this, we established a model of recovery from a single sub-lethal dose of TAA, and compared the FVB, C57Bl/6, and BALB/c strains' responses 1, 3, 5, and 8 days thereafter (Fig. 1a). Biochemical quantification of hepatic collagen revealed readily detectable progressive collagen deposition into BALB/c livers starting from day 3 and reaching maximum (142% above control group levels, $p < 0.001$) on day 8 post TAA. In contrast, both FVB and C57Bl/6 mice developed significantly less fibrosis, with maximum collagen levels reaching 77% and 88%, respectively, above their healthy controls (Fig. 1b). Upon histological examination, connective tissue deposits were readily detected by Sirius Red staining starting on day 8 in pericentral areas of BALB/c mice, but not of FVB or C57Bl/6 mice (Fig. 1d). The degree of hepatotoxin-induced injury was similar between all three strains studied, as measured by cumulative serum alanine aminotransferase (ALT) levels 12–96 h post TAA ("area under curve" analysis, Fig. 1c). Overall, based on data in this single-dose TAA injury model, we were able to recapitulate strain differences in susceptibility to fibrosis observed in the chronic TAA injury model (BALB/c > C57Bl6 > FVB, Supplementary Fig. 1). Importantly, similar results were obtained in FVB and BALB/c mice after acute injury with another hepatotoxin, carbon tetrachloride (CCl4), suggesting that these strain differences were not toxin specific (Supplementary Fig. 2).

**Delayed dead hepatocyte clearance precedes fibrotic response.** Further histological examination revealed that by 24 h, TAA caused a similar degree of pericentral hepatocyte cell death in ~50% of hepatocytes in each strain. However, striking differences were observed in the speed with which the necrotic masses were cleared. In the resistant FVB strain, necrotic hepatocytes were rapidly cleared and disappeared completely by day 3, while in BALB/c they persisted even after 5 days (Fig. 2a).

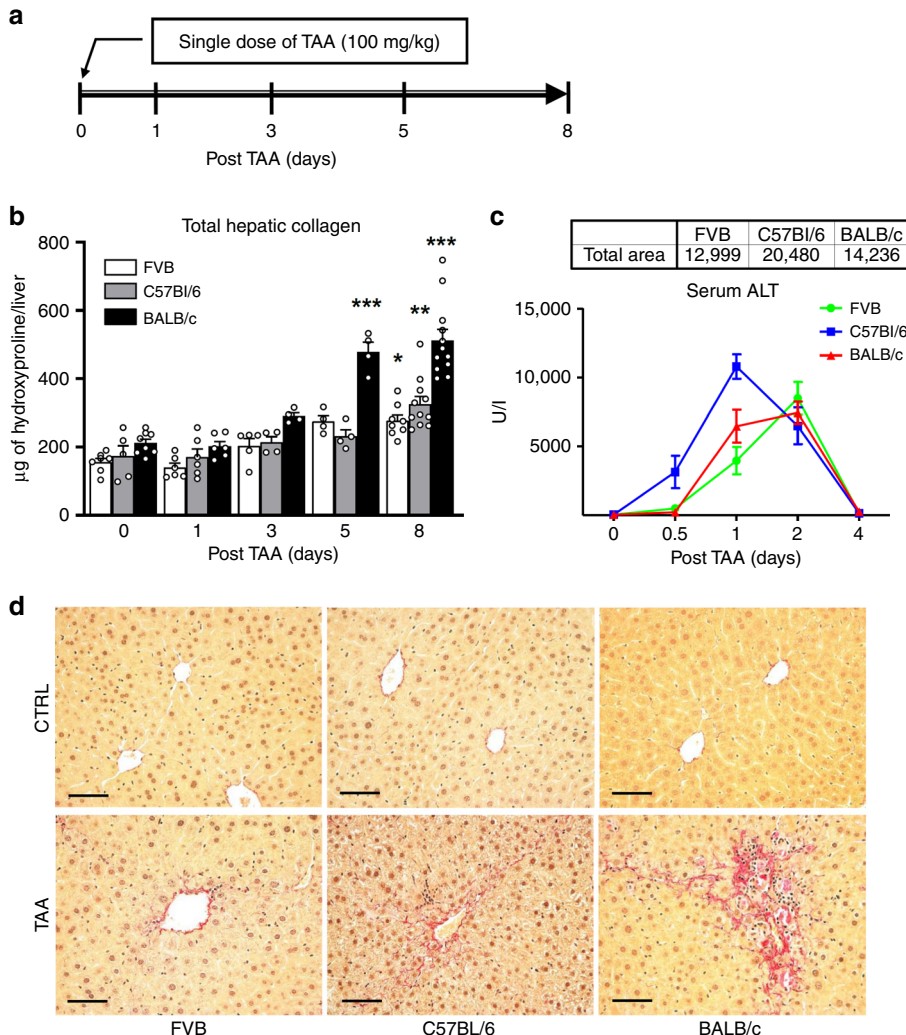

**Fig. 1 Fibrotic responses to acute TAA-induced hepatocyte injury recapitulate strain differences in fibrosis susceptibility (BALB/c > C57Bl/6 > FVB) in chronic liver fibrosis model. a** Experimental design of acute TAA-induced incipient liver fibrosis model in mice. Injury and wound healing responses were evaluated at 1, 3, 5, and 8 days after a single, acute sub-lethal dose of TAA (100 mg/kg, i.p.). **b** Significant collagen synthesis and deposition is readily detected in fibrosis-susceptible BALB/c mice after 5 days post-TAA injection by total hepatic collagen content measurements, with 2.5-fold increase in BALB/c strain, but not in FVB strain, after 5 days. For each strain at 0/1/3/5/8 days time-points, $n = 7/6/5/4/8$ (FVB), $n = 5/6/4/4/11$ (C57Bl/6), $n = 8/6/4/4/12$ (BALB/c) of individual animals. *$P < 0.05$; **$p < 0.01$; and ***$p < 0.001$ compared to respective strain controls (one-way ANOVA, followed by Tukey's post test). **c** Extent of liver injury in response to TAA is comparable among studied strains and does not determine susceptibility to fibrosis. Cumulative hepatotoxicity of a single TAA injection was determined by serum ALT levels at early time-points at 12, 24, 48, and 96 h, which were used to calculate area under the curve (AUC). For each strain at 0.5/1/2/4 days time-points, $n = 4/6/9/4/11$ (FVB), $n = 4/6/6/6/7$ (C57Bl/6), $n = 4/6/6/6/10$ (BALB/c) of individual animals. AUC was similar between FVB and BALB/c strain (see total area under curve values in the table above the graph) and was greater in C57Bl/6, demonstrating no direct relationship of TAA hepatotoxicity with fibrosis susceptibility. **d** Connective tissue staining demonstrates scarless repair of pericentral areas in FVB, but pronounced pericentral fibrosis in BALB/c strain. C57Bl/6 mice show modest fresh fibers deposition. Representative images of pericentral areas in livers 8 days post TAA is shown (Sirius Red, original magnification, ×200; bar, 50 μm). C57Bl/6 strain demonstrates intermediate fibrotic response. Total HYP (mg/whole liver) was calculated from individual liver weights and respective relative HYP values ($n = 4$–12). Ctrl: non-fibrotic control group ($n = 4$) that received vehicle (saline) only; TAA: fibrotic mice treated with single-dose TAA ($n = 4$–12). Data are expressed as means ± SEM. Source data are provided as a Source Data file.

C57Bl/6 strain demonstrated necrotic masses clearance rate similar to fibrosis-resistant FVB with a delay of about 24 h. All subsequent experiments were performed in FVB and BALB/c mice, which demonstrated the most divergent responses. Macroscopically, at day 8 of recovery from TAA injury, BALB/c livers demonstrated increased stiffness and patchy whitish lesions throughout the liver parenchyma, while FVB livers appeared healthy and virtually indistinguishable from untreated controls (Fig. 2b). In both strains, recovery from TAA injury was accompanied by a robust up-regulation of pro-fibrogenic messenger RNA (mRNA) encoding for TGFβ1, pro-collagen α1 (I), TIMP-1, and matrix metalloproteinase-2 (MMP-2), with the peak at day 3, except for TIMP-1, which was induced after 1 day (Fig. 2c). Surprisingly, the degree of transcript up-regulation was nearly identical between the two strains in the early phase (0–3 days), but differed significantly only at the late time-points (5 and 8 days). After day 5, most pro-fibrogenic mRNAs normalized in FVB mice, but remained elevated in BALB/c, with the strain differences most apparent at day 8 post injury (Fig. 2c).

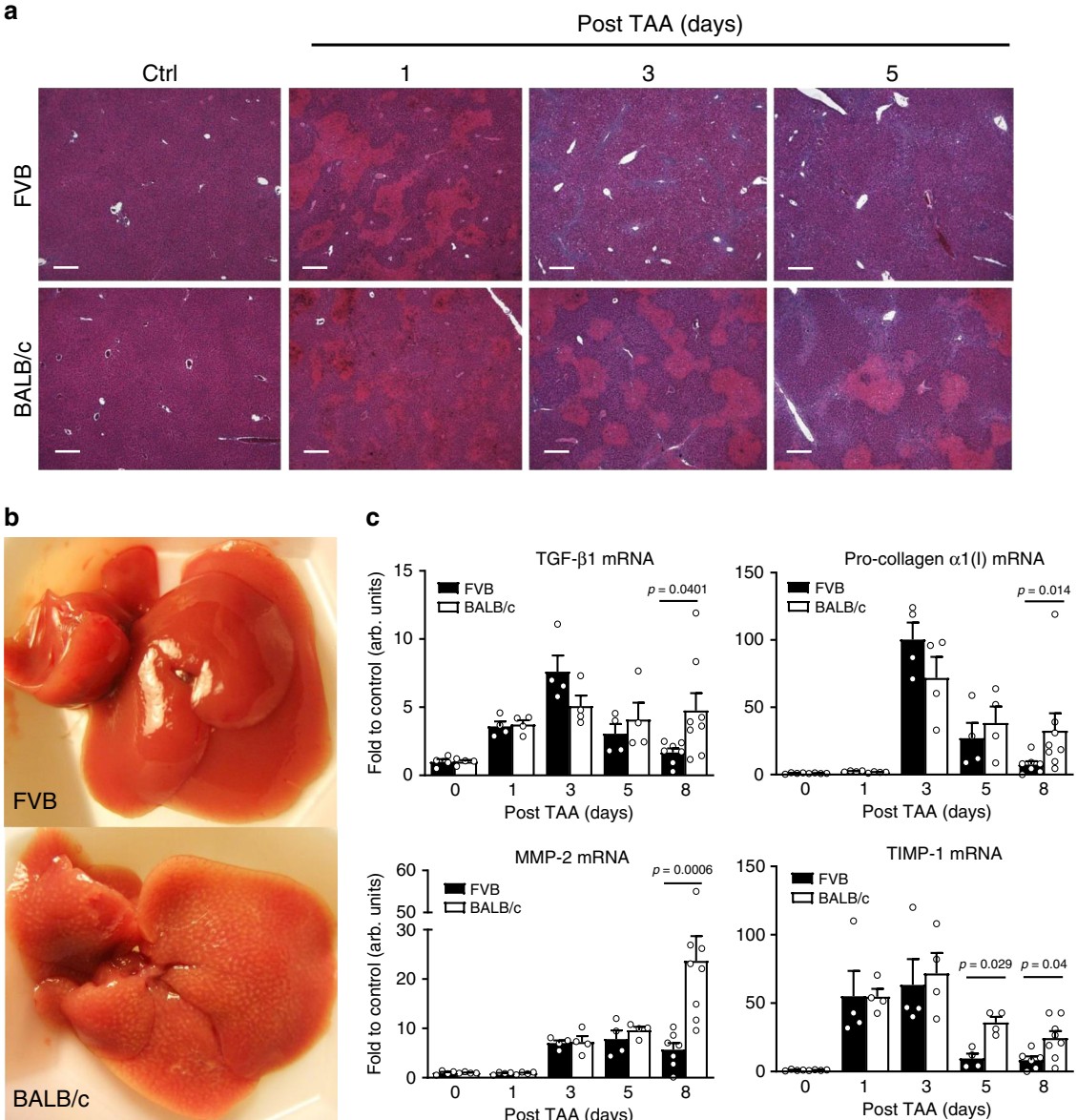

**Fig. 2 Delayed clearance of dead hepatocytes after injury precedes amplified fibrotic response in fibrosis-susceptible BALB/c strain. a** Histological examination suggests that single-dose TAA causes similar cell death in livers of FVB and BALB/c mice, but that the subsequent clearance rate of necrotic masses differs significantly. Necrotic masses (eosinophilic areas) are rapidly cleared in fibrosis-resistant FVB mice (by day 3), but persist in fibrosis-susceptible BALB/c mice past day 5 (representative low-magnification images, hematoxylin/eosin, ×50; bar, 20 μm). **b** Macroscopic appearance of representative livers from FVB and BALB/c mice 8 days post-TAA injury. FVB livers completely recovered by day 8 and appear virtually indistinguishable from healthy uninjured liver, while BALB/c livers demonstrate increased turgor, stiff consistence, and strong whitish pattern indicative of incomplete repair. **c** Pro-fibrogenic gene expression in post-TAA livers indicate that delayed normalization of multiple pro-fibrogenic mRNA expression is associated with fibrosis susceptibility in BALB/c mice. No significant strain difference at peak of fibrogenic response was observed (day 3). Hepatic expression of pro-fibrogenic (pro-collagen α1(I), TGFβ1, TIMP-1, and MMP-2) transcript levels was quantified by QRT-PCR during recovery (0–8 days) after single TAA injury. Results are expressed as means ± SEM, fold to healthy wild-type controls relative to β2MG mRNA (for each strain at 0/1/3/5/8 days time-points, $n = 4/4/4/4/7$ (FVB), $n = 4/4/4/4/8$ (BALB/c) of individual animals). *P* value as indicated (two-tailed, Mann–Whitney *t* test, not adjusted) when both strains are compared at corresponding time-point. Source data are provided as a Source Data file.

**Impaired efferocytosis function in fibrosis-susceptible mice.** The major cells responsible for dead cell recognition and clearance in vivo are macrophages ("professional phagocytes"), which prompted us to investigate temporo-spatial changes in macrophage infiltrate. Both strains demonstrated similar immune cell recruitment in early injury (12–24 h post-TAA) stages, as assessed via hepatic mRNA expression of pan-leukocyte marker CD45 and key macrophage/monocyte markers F4/80, Clec4F, CD68, and CD11b/c (Supplementary Fig. 3a). However, notable temporo-

spatial differences became apparent in the intermediate (3 days) and late (8 days) repair phase. In situ F4/80 staining showed that MΦ infiltrated necrotic areas at day 3 in FVB mice, and these infiltrates cleared completely by day 8. In contrast, in BALB/c mice macrophages accumulated at the edge of necrotic areas at day 3, and remained in large numbers even at day 8, suggesting incomplete efferocytosis (Supplementary Fig. 3b). Immune cell infiltrates in both strains also stained positive for myeloid cell markers Gr-1 (monocytes and granulocytes) and CD11b

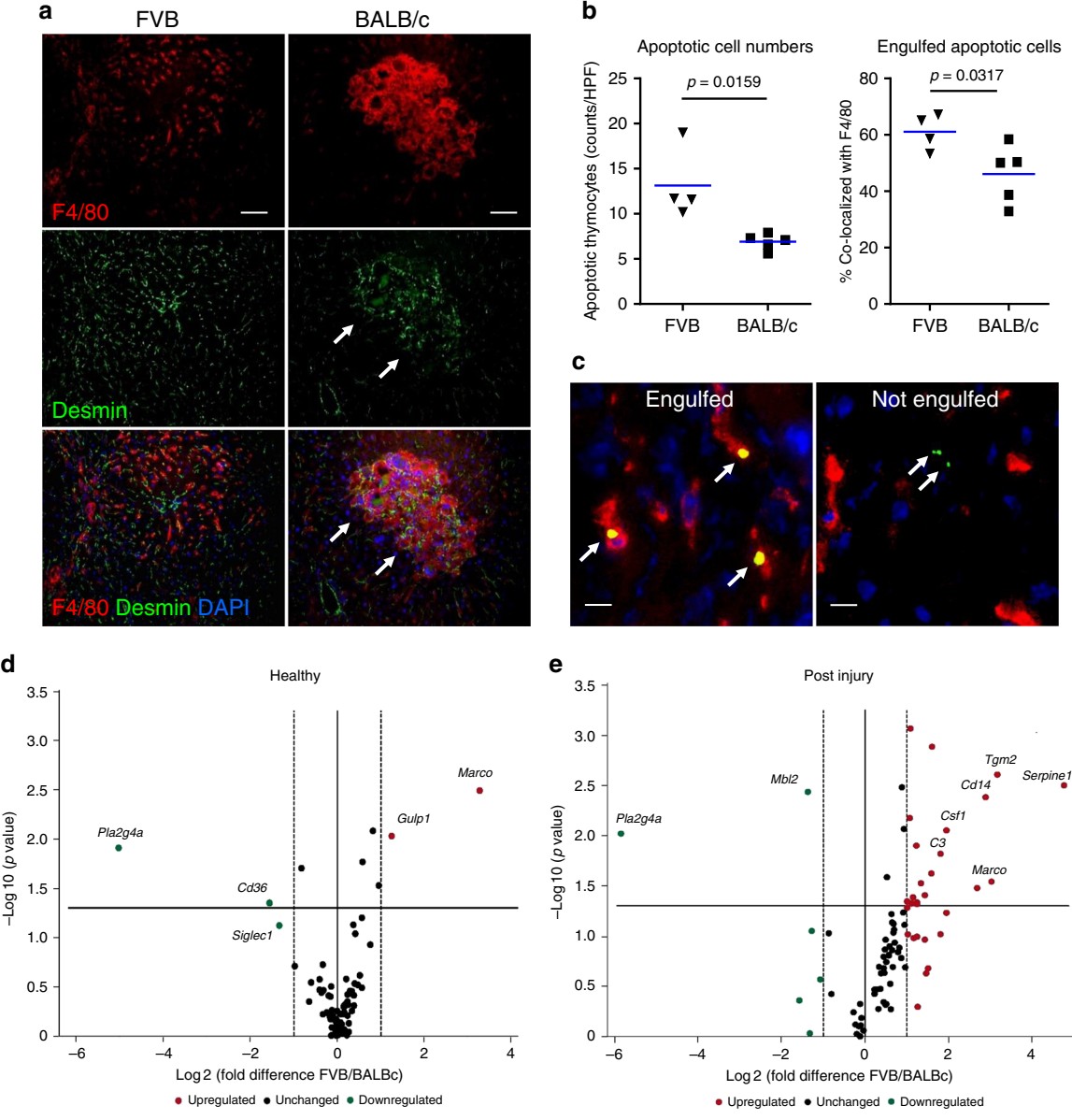

**Fig. 3 Impaired phagocytic macrophage function and efferocytosis after liver injury in fibrosis-susceptible BALB/c strain. a** Double immunofluorescence staining for the HSC marker desmin (green) and macrophage cell marker F4/80 (red) in FVB (left column) and BALB/c mice (right column) in late stage (8 days) of recovery. Normal peri-sinusoidal localization of desmin-positive stellate cells and F4/80-positive macrophages is restored in FVB mice, but not in BALB/c mice. Note large clusters of desmin-positive stellate cells (arrow) accumulate within granuloma-like macrophage infiltrates persisting in pericentral areas of BALB/c livers. Representative images at original magnification (×200; bar, 50 μm). **b** Total apoptotic cell counts (left panel) and % of apoptotic thymocytes engulfed by hepatic F4/80+ macrophages (right panel) significantly increased in the livers of FVB versus BALB/c mice 48 h post-TAA-induced liver injury. In vivo apoptotic thymocytes phagocytosis assay was performed as described in "Methods" and analyzed by cell scoring/counting in 10 random HPF per mouse. n = 4 (FVB), n = 5 (BALB/c); p value as indicated (two-tailed, Mann–Whitney t test). **c** Representative images of apoptotic cells (green, arrows; bar, 10 μm) scored as engulfed (left panel) or non-engulfed (right panel) by hepatic F4/80+ macrophages (red) 1 h after i.v. infusion of fluorescently labeled apoptotic thymocytes (green). **d, e** Volcano plots comparing statistically significant phagocytosis-related gene expression changes between FVB and BALB/c strains in total RNA of naive (**d**, healthy) or injured (**e**, 48 h post-TAA) livers. Total of 84 phagocytosis genes analyzed via RT²PCR phagocytosis array; values are obtained from four biological replicates after normalization to housekeeping gene β2MG. Log 2 values of the fold changes are plotted on the x-axis, the −log 10-transformed p values are plotted on the y-axis. The solid line in the graphs marks the p value of 0.05, and the dotted line marks a one-fold change. While only few genes were differentially expressed at baseline (C, two genes upregulated and two genes down-regulated), 20 genes were upregulated in response to injury in FVB mice compared to BALB/c (p < 0.05, t test). Differentially regulated genes by mini-array are annotated, and subset of genes cross-validated by TaqMan RT-PCR method are shown in bold (see also Supplementary Tables 4–6 for complete list of genes and TaqMan RT-PCR validation data). Source data are provided as a Source Data file.

(monocytes) (Supplementary Fig. 3c, d). Double immunostaining for F4/80 and desmin revealed that active fibrogenesis likely occurs within these persistent macrophage infiltrates in BALB/c mice, as evidenced by marked accumulation of desmin+ HSCs/MFs; in contrast, desmin+ HSCs/MFs retained normal peri-sinusoidal distribution in resistant FVB strain (Fig. 3a).

In order to directly assess whether function of dead cell removal by phagocytes (efferocytosis) is indeed impaired in the livers of the fibrosis-susceptible strain, we performed an in vivo phagocytosis assay via administration of strain-matched fluorescently labeled apoptotic thymocytes into FVB and BALB/c mice (both healthy and after TAA-induced liver injury). We then measured their engulfment (localization within macrophages) 1 h later. Analysis of apoptotic cell distribution in relation to macrophage markers revealed that liver homing of apoptotic cells, as well as their engulfment rate by F480+-resident macrophages, was significantly lower in injured livers of BALB/c mice compared to FVB (Fig. 3b, c). Apoptotic cell numbers were similar in livers of healthy FVB and BALB/c, and demonstrated similarly high co-localization with F4/80+ macrophages (>85% engulfed). After liver injury, apoptotic cell homing and percent of engulfed apoptotic cells were depressed in both strains; however, the magnitude of change was remarkably more pronounced in BALB/c compared to FVB. Thus, total apoptotic cell counts decreased by 57% (BALB/c) versus 31% (FVB), whereas percent of apoptotic cell engulfed by macrophages decreased by 43.5% (BALB/c) versus 24.7% (FVB) compared to healthy strain-matched controls (Supplementary Table 3). Proportion of apoptotic cells co-localized with infiltrating Gr-1+ myeloid cells in injured livers was very low (>5%) compared to that of F4/80+ macrophages, and did not differ between strains (Supplementary Table 3).

Finally, to identify phagocytic genes potentially responsible for inefficient efferocytosis in the fibrosis-susceptible BALB/c strain, we profiled hepatic expression of phagocytosis-related genes using mini-array. Among the 84 relevant genes analyzed, only four genes were differentially expressed between strains at baseline (Fig. 3d). In contrast, 48 h after injury, 20 genes were upregulated in response to injury in FVB mice compared to BALB/c (Fig. 3e, $p < 0.05$, see Supplementary Tables 4 and 5 for a complete list of genes). Upon additional cross-validation using TaqMan PCR for the top 10 differentially regulated genes, five genes were confirmed as significantly different between strains post injury (Supplementary Table 6). Among these were genes encoding phagocytic receptors (Cd14, Marco); recognition and engulfment molecules (Csf1); and phagosome maturation (Serpine1, Tgm2). Thus, delayed clearance of necrotic masses (Fig. 2a), inefficient efferocytosis (Fig. 3b, c), and persistence of macrophage infiltrates (Supplementary Fig. 3) in the injured livers of fibrosis-susceptible BALB/c mice associated with failure to efficiently upregulate multiple genes involved in the phagocytic pathway.

**Phagocyte depletion abrogates resistance to fibrosis.** To clarify the mechanistic role of phagocytic liver macrophages and infiltrating myeloid cells in fibrosis susceptibility, we used both injection of clodronate-loaded liposomes (CLO), which efficiently eliminates tissue macrophages with high phagocytic activity (Supplementary Fig. 4a), and antibody-mediated depletion of myeloid cell subsets. First, we depleted the macrophages either 1 day before or 1 day after TAA-induced injury by a single intraperitoneal (i.p.) injection of clodronate-containing liposomes, to eliminate (1) resident phagocytic macrophages or (2) resident and recruited/infiltrating phagocytic macrophages, respectively (Fig. 4a) and compared fibrotic responses at the late

(8 days post injury) time-point. Based on quantitative analysis of collagen deposition in these animals, macrophage depletion in resistant FVB mice rendered them completely susceptible to fibrosis, and caused only minor changes in collagen deposition in BALB/c mice (Fig. 4b). Histologic examination revealed that MΦ-depleted FVB mice failed to clear necrotic masses of hepatocytes, which persisted at day 8 post injury (Supplementary Fig. 4b) and developed fibrotic lesions histologically similar to that observed in fibrosis-susceptible BALB/c strain, regardless whether depletion was performed before or after injury (Fig. 4c). This was accompanied by a dramatic increase in pro-fibrogenic gene expression: TGFβ1, pro-collagen α1(I), and TIMP-1 were upregulated 2–6-folds above controls in FVB mice and did not normalize by day 8 in both strains with MΦ depletion (Fig. 4d). Upon depletion of phagocytes, there was virtually no difference in fibrotic response between these two mouse strains upon assessment by histology or quantitative fibrosis measures (Fig. 4). Importantly, we were able to reproduce these results when chronic MΦ depletion was performed concurrently with chronic TAA administration for 6 weeks. Resistance to fibrosis was compromised in FVB mice receiving weekly CLO injections, which developed significant hepatic fibrosis, as evidenced by increase in collagen deposition, histological signs of bridging fibrosis, and increase in HSC activation marker α-SMA, compared to controls treated with TAA and vehicle [phosphate-buffered saline (PBS)]-loaded liposomes (Supplementary Fig. 5). Since both resident macrophage and infiltrating Gr-1(+)/CD11b(+) myeloid cells appeared to accumulate within post-necrotic lesions (Fig. 3, Supplementary Fig. 3), we also performed myeloid cell subset depletions using lineage-specific antibody during recovery of resistant FVB mice from TAA injury (Fig. 4a). Cell depletion using Gr-1 (but not CD11b or Ly-6G)-specific antibody resulted in persistence of dead hepatocyte at day 8 post injury, and compromised fibrosis resistance in FVB mice (Fig. 4e). Of note, administration of Gr-1 (but not CD11b or Ly-6G)-specific mAB led to HSC activation on day 8 post TAA (Supplementary Fig. 6a) and increased hepatic collagen deposition as determined biochemically via hydroxyproline (Supplementary Fig. 6b). These data collectively suggest that both resident liver macrophages [Kupffer cells (KCs)] and infiltrating Gr-1(+) myeloid cell subset are functionally required for efficient clearance of dead hepatocytes, and appear to protect from ensuing fibrotic responses.

**Hepatocyte-derived mito-DAMPs compromise fibrosis resistance.** The results described above suggested to us that prolonged exposure to the certain DAMPs released from persisting dead hepatocytes in fibrosis-susceptible strain (or when MΦ function is compromised in resistant FVB mice) may be directly responsible for triggering exaggerated fibrotic responses. One such group of molecules leaking from damaged mitochondria (collectively called mito-DAMPs) is particularly immunogenic and has been implicated in various diseases[8]. Indeed, we detected transient 3-fold elevation in serum levels of mtDNA (a marker of mito-DAMPs release) 2 days post-TAA injury in fibrosis-susceptible BALB/c mice, but not in resistant FVB mice (Fig. 5a). Importantly, this was not due to strain differences in mtDNA genome copy number, which was similar in FVB and BALB/c livers relative to nuclear genome (Supplementary Fig. 7d, e). To test if mito-DAMPs were directly pro-fibrogenic, we purified liver mitochondria, prepared mito-DAMPs from them, and injected them into resistant FVB mice on the second and fourth day post-TAA injury to mimic prolonged exposure to mito-DAMPs due to inefficient efferocytosis of damaged hepatocyte in vivo (Fig. 5b). The mito-DAMP dose chosen contained 9.5 μg of mtDNA, which corresponds to 10% of the whole liver and in all likelihood reflects

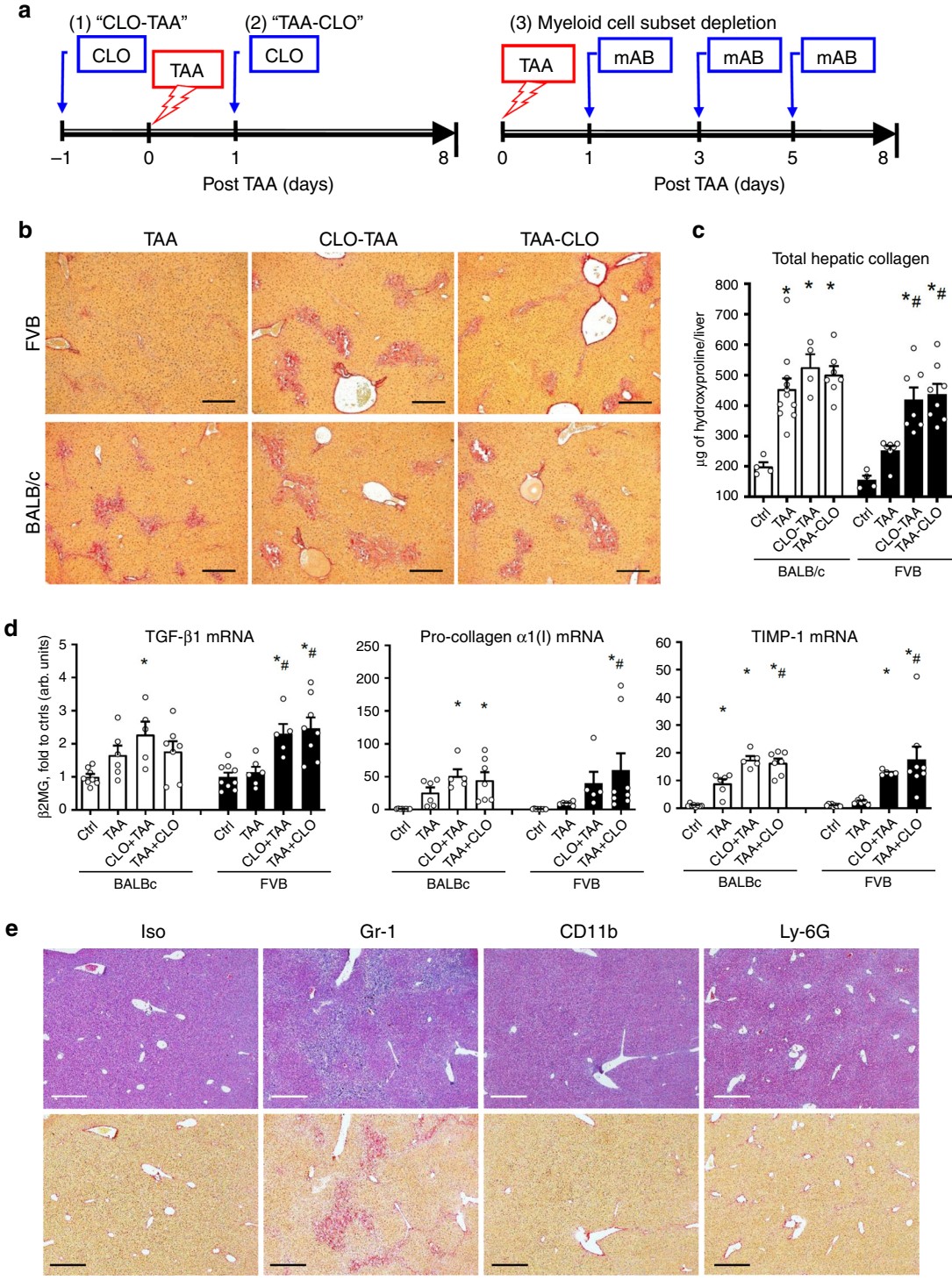

pathophysiologic levels based on 40–50% hepatocyte death observed in response to TAA in our system and kinetics of systemic exogenous mito-DAMPs exposure in vivo (Supplementary Fig. 7c). While control FVB mice, as expected, were able to repair the liver injury in a scarless manner by day 8 post injury, FVB mice injected with mito-DAMPs developed significant liver fibrosis, with readily detectable collagen deposits, morphologically characterized as incomplete fibrotic septa (Fig. 5c). Quantitatively, the overall increase in collagen deposition in mice injected with mito-DAMPs was significant, with a 2.5-fold increase as determined via connective tissue square morphometry, and a 27.45% increase as determined via biochemical hydroxyproline

assay, respectively ($p < 0.05$, Fig. 5d, e). Areas of collagen deposition in livers of mito-DAMP-treated mice overlapped with a massive increase in immunopositivity for α-SMA, an HSC activation marker. Moreover, pro-fibrogenic gene transcript levels of TGFβ1, pro-collagen α1(I), and TIMP-1 in livers of mito-DAMP-treated mice remained elevated 3–4-folds at day 8 post injury (Fig. 5f). In order to validate these findings in a second, mechanistically different model of steatohepatitis, we injected mice primed with short-term methionine- and choline-deficient diet (MCD) feeding (2 weeks) with a single dose of purified mito-DAMPs (9.5 µg/mouse, i.p.) and examined the livers 24 h later; in situ immunostaining for α-SMA revealed massive activation of

**Fig. 4 Phagocytes depletion amplifies fibrogenic response and abrogates resistance to fibrosis in FVB mice, blunting strain differences after acute liver injury. a** Scheme of phagocytic macrophage depletion experiment where single injection of clodronate liposomes was performed either (1) 24 h prior (CLO-TAA) or (2) 24 h after (TAA-CLO) TAA administration, and (3) myeloid cell subsets depletion via administration of Gr-1, CD11b, or Ly-6G-specific mAB on first, third, and fifth day post TAA. All mice were sacrificed and evaluated at day 8 post TAA. **b** Connective tissue staining demonstrates significant pericentral deposition of collagen in FVB mice with macrophage depletion, virtually indistinguishable from lesions in BALB/c. No collagen deposition occurs in FVB mice without macrophage depletion (×50; bar, 100 μm). **c** Collagen deposition in livers of FVB and BALB/c mice with macrophage depletion before or after liver insult (for Ctrl/TAA/CLO + TAA/TAA + CLO groups, $n = 4/11/4/7$ (BALB/c), $n = 4/7/7/8$ (FVB) of individual animals). Macrophage depletion (before or after injury) abrogates fibrosis resistance in FVB mice, while having a minor effect on collagen levels in fibrosis-susceptible BALB/c mice. **d** Pro-fibrogenic gene expression in post-TAA livers demonstrates that fibrogenic responses are amplified and persist longer in FVB mice treated with CLO, similarly to levels observed in fibrosis-susceptible BALB/c. Hepatic expression of pro-fibrogenic (TGFβ1, pro-collagen α1(I), and TIMP-1) transcript levels was quantified by QRT-PCR 8 days after single TAA injury. Results are expressed as means ± SEM, and in arbitrary units (fold to healthy wild-type controls) relative to β2MG mRNA as described in "Methods." **e** Fibrosis-resistant FVB mice were administered cell subset-specific antibody to deplete myeloid cells (Gr-1 mAB RB6-8C5, 200 μg/mouse), monocytes (CD11b mAB M1/70, 200 μg/mouse), isotype control (Iso, LTF-2 IgG2b, 200 μg/mouse), or granulocytes/neutrophils (Ly-6G mAB 1A8, 500 μg/mouse) on first, third, and fifth day after TAA injections (for Ctrl/TAA/CLO + TAA/TAA + CLO groups, $n = 8/6/5/7$ (BALB/c) and $n = 9/6/5/8$ (FVB) of individual animals). Representative low-magnification (×50; bar, 100 μm) images show serial liver sections stained with H/E (upper row) and connective tissue staining (lower row, picrosirius red). *$P < 0.05$ compared to healthy controls of respective strain; #$p < 0.05$ compared to TAA group of respective strain without macrophage depletion (ANOVA with Dunnett's post test). Source data are provided as a Source Data file.

**Table 1 Characteristics of the healthy human subjects and treatment-naive patients with biopsy-proven NAFLD/NASH.**

|  | Healthy | NASH (cohort I) | NASH (cohort II) |
|---|---|---|---|
| Total number | 12 | 27 | 114 |
| Gender, male/female | 9/3 | 14/13 | 72/42 |
| Age, mean ± SD (range) | 42.2 ± 12.4 (25–63) | 52.4 ± 12.9 (21–69) | 56.4 ± 12.3 (30–90) |
| NAS, 0–3/4–8 | – | 7/20 | 24/90 |
| Fibrosis, F0–1/F2–4 (Brunt/Kleiner score) | – | 13/14 | 68/46 |

peri-sinusoidal HSCs throughout the liver lobule, with a 2.6-fold increase in α-SMA area morphometrically (Fig. 5g).

**Mito-DAMPs directly trigger pro-fibrogenic HSC activation.** In order to investigate whether pro-fibrotic action of mito-DAMPs may be due to direct effect on HSCs, a main fibrogenic effector cell in the liver, we freshly isolated HSCs and incubated them with increasing concentrations of mito-DAMPs (0.25, 0.5, and 0.75 μg/ml) in vitro. Twenty four hours after the addition of mito-DAMPs, HSCs demonstrated dose-dependent changes in morphology characteristic of their activation, with the loss of lipid droplets, assumption of an MF-like appearance, and robust increases in immunopositivity for α-SMA (Fig. 6a). HSC proliferation, as assessed by MTT [3-(4,5-dimethylthiazol-2-yl)- 2,5-diphenyltetrazoliumbromide] assay, was dose-dependently increased in the presence of exogenously added mito-DAMPs (Fig. 6b). Concomitantly, primary HSCs upregulated pro-fibrogenic gene expression of pro-collagen α1(I), TGFβ1, and TIMP-1 (2–4-folds over controls, Fig. 6c). Pre-incubation of mito-DAMPs with DNase I immediately prior to the addition to HSC cultures efficiently depleted mtDNA and significantly attenuated its pro-fibrogenic activity by up to 50% compared to intact mito-DAMP preparations, as assessed via cell proliferation, α-SMA induction, and fibrogenic gene expression (Fig. 6d–h). Interestingly, primary resident liver macrophages (Kupffer) cells failed to respond to equivalent doses of mito-DAMPs in vitro, both at baseline and upon activation by lipopolysaccharide (LPS) or interferon-γ (IFNγ), as measured by secretion of tumor necrosis factor-α (TNF-α), interleukin-1β (IL-1β), or nitric oxide (NO) (Supplementary Fig. 8). Taken together, these results

suggest that exposure to exogenous mito-DAMPs directly triggers fibrogenic activation in HSCs, with mtDNA as a major active component, as evidenced by characteristic changes in morphology, up-regulation of α-SMA, increased cell proliferation, and pro-fibrogenic transcriptional activity.

**Circulating mito-DAMPs are elevated in human NAFLD/NASH.** In order to test if circulating mito-DAMPs levels are increased in human disease, we performed a survey of mtDNA levels in the serum of 27 patients (cohort I) with biopsy-confirmed non-alcoholic fatty liver disease (NAFLD) via PCR amplification of a short fragment in the D-loop region of mtDNA. The majority of patients (20/27) in this cohort had active NASH, defined histopathologically as NAFLD activity score (NAS) ≥ 4, and various degrees of fibrosis ranging from F0 (no fibrosis) to F4 (cirrhosis). Twelve healthy controls of different ages (24–63 years old) were analyzed for comparison (Table 1). Serum mtDNA levels were increased 7-fold on average in NAFLD patients with different stages of disease when compared to healthy subjects ($p < 0.0001$), with mtDNA levels reaching up to a 17-fold increase above average normal values (Fig. 7a). Circulating mtDNA values in healthy subjects did not appear to vary with age or sex. Association with histological parameters of disease progression (NAS score and fibrosis stage on liver biopsy) was tested in an additional 114 patients with NAFLD/NASH (cohort II). Circulating mtDNA levels were further increased in patients with active NASH (NAS score 4–8, versus minimal disease activity NAS score 0–3, $p = 0.0334$, Fig. 7b) and particularly in those with the significant histological signs of fibrosis [F2–4, 482.4 ± 62.66, 95% confidence interval (CI), 356.2–608.6] on biopsy, compared to patients with minimal/no fibrosis (F0–1, 253.4 ± 26.45, 95% CI, 200.6–306.2, $p = 0.0003$, Fig. 7c).

**Discussion**
It is well recognized that individuals with chronic liver disease exhibit profound differences in fibrosis progression rates. This poorly understood clinical phenomenon can be replicated in genetically divergent inbred mouse strains that demonstrate various degrees of susceptibility to experimental liver fibrosis. Studies in inbred mice may potentially provide valuable mechanistic insights into fibrosis susceptibility in humans, and help to devise new antifibrotic therapies or identify patients at high risk of progression.

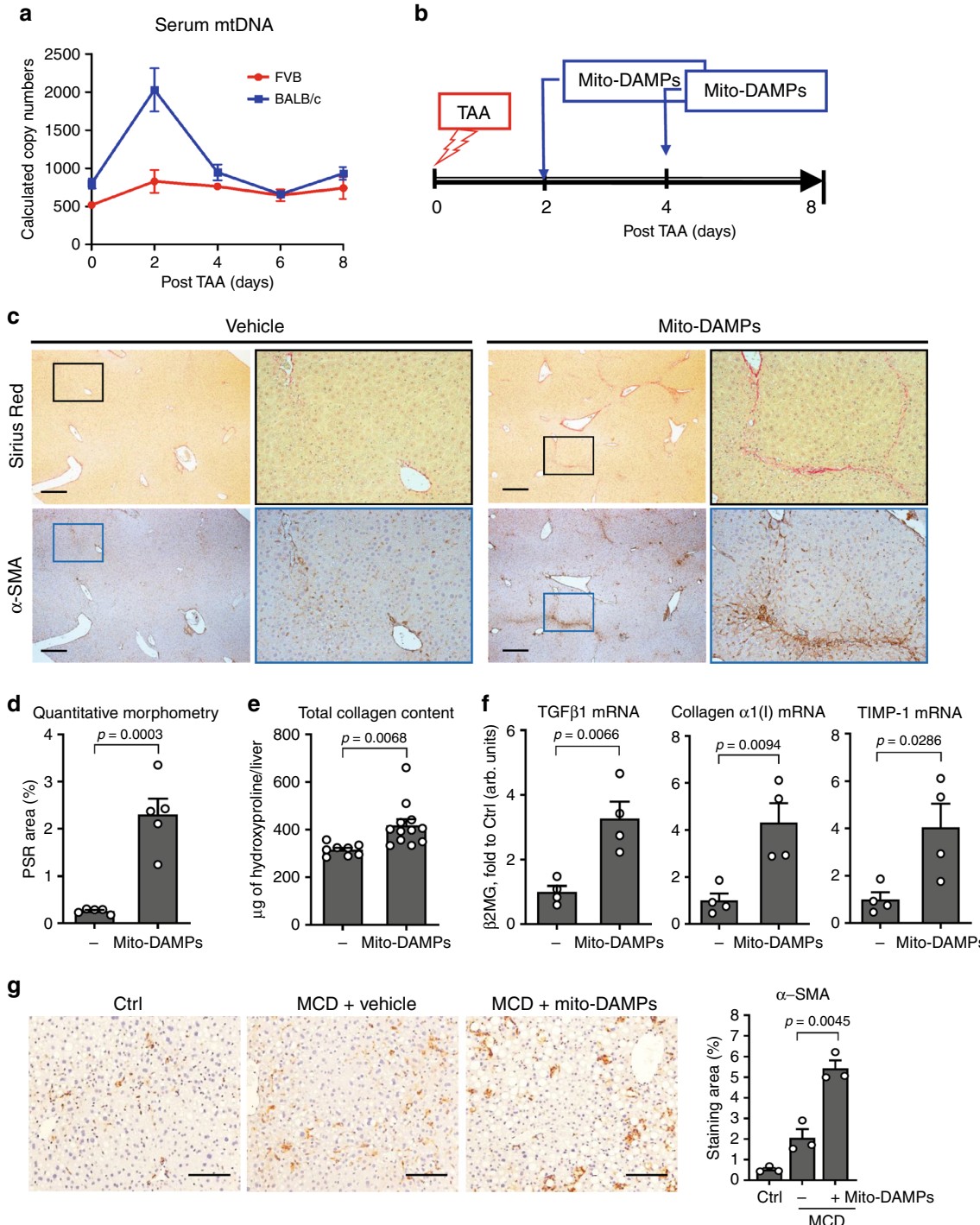

**Fig. 5 Administration of hepatocyte-derived mito-DAMPs amplifies fibrogenic HSC activation in vivo and circumvent the resistance to fibrosis in FVB mice with TAA-induced liver injury. a** Release of mito-DAMPs into systemic circulation is elevated in fibrosis-susceptible BALB/c, but not in resistant FVB strain 2 days after single TAA-induced liver injury, as quantified by mtDNA levels in the serum. Data shown are 12S mtDNA region QRT-PCR expressed as target copy number/μl of serum (means ± SEM, for 0/2/4/6/8 days time-points, $n = 6/5/4/9/4$ (FVB), and $n = 6/3/5/10/5$ (BALB/c) of individual animals). **b** Scheme of exogenous hepatocyte mito-DAMP administration experiment. Purified mito-DAMPs (9.5 μg of mtDNA/mouse) were injected intraperitoneally into FVB mice 2 and 4 days post-TAA injury, and fibrotic responses evaluated 8 days after TAA. **c** Connective tissue staining (Sirius Red, upper panel) and immunohistochemistry for α-SMA (lower panel) in livers of FVB mice with mito-DAMP injection. Representative images shown at low (×50; bar, 50 μm) and high (×200) magnification as indicated. **d** Quantitative morphometry of collagen area ($n = 5$). **e** Hepatic collagen deposition is increased in mito-DAMP-treated FVB mice, as assessed biochemically via hydroxyproline content (controls (−), $n = 8$; mito-DAMPs, $n = 12$ individual animals). **f** Hepatic expression of pro-fibrogenic (pro-collagen α1(I), TGFβ1, and TIMP-1) transcript levels as quantified by QRT-PCR ($n = 4$). **g** Representative α-SMA staining images in C57Bl/6 mice with early-stage steatohepatitis due to MCD feeding for 2 weeks with morphometric quantification (×100, $n = 3$; bar, 50 μm), 24 h after mito-DAMP administration (9.5 μg of mtDNA/mouse, i.p.). Results are expressed as means ± SEM, and in arbitrary units (fold to healthy controls) relative to β2MG mRNA as described in "Methods." Two-tailed $p$ value indicated as compared to vehicle controls (unpaired $t$ test). Source data are provided as a Source Data file.

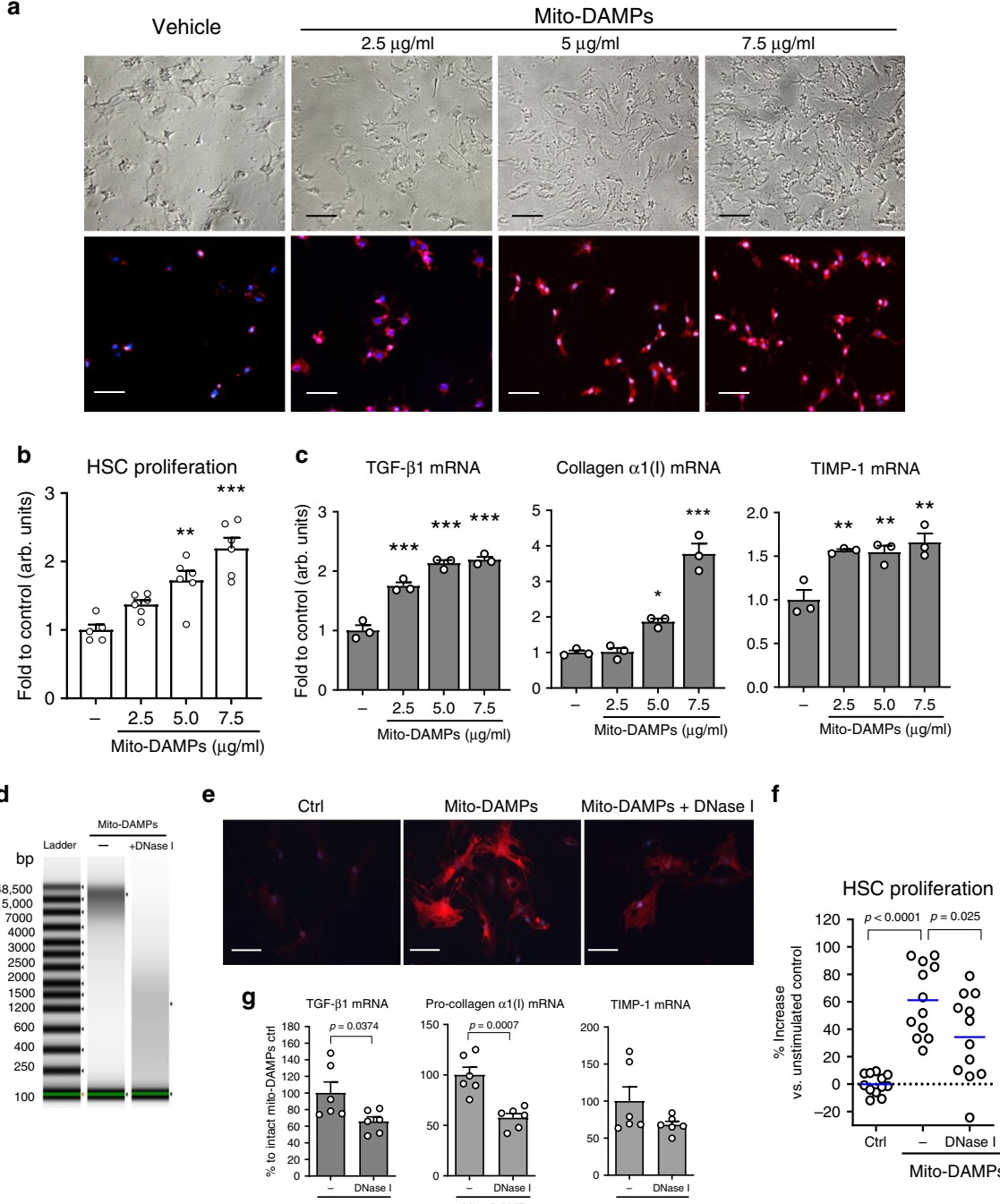

**Fig. 6 Hepatocyte-derived mito-DAMPs directly trigger fibrogenic activation of primary hepatic stellate cells, with mtDNA as a major active component.** Freshly isolated murine HSCs were incubated for 24 h in vitro with increasing concentration of mito-DAMPs (corresponding to 2.5–7.5 µg of intact mtDNA/ml) prepared from purified liver mitochondria. **a** Changes in HSC morphology upon mito-DAMPs exposure (upper panel, phase contrast and lower panel, immunofluorescence for HSC activation marker α-SMA; bar, 50 µm). Representative images of two independent experiments (cell isolations). **b** HSC proliferation assessed by the MTT assay ($n = 5$ of biological replicates for Ctrl and $n = 6$ for mito-DAMP-treated groups, results are representative of two independent experiments with similar results). **c** Pro-fibrogenic gene expression of pro-collagen α1(I), TGFβ1, and TIMP-1 quantified by QRT-PCR ($n = 3$ of biological replicates, results are representative of two independent experiments with similar results). **d** DNA microgel image showing complete mtDNA degradation following pre-incubation with DNase I as described in "Methods." Fibrogenic activation of HSCs by mito-DAMPs (7.5 µg/ml) is significantly attenuated by mtDNA depletion, as assessed via **e** α-SMA immunofluorescence (bar, 10 µm; images are representative of two independent experiments), **f** cell proliferation (MTT assay, blue line indicates mean values, $p$ values as indicated via ANOVA with Tukey's post test, $n = 12$ biological replicates derived from two independent experiments), and **g** pro-fibrogenic gene expression of pro-collagen α1(I), TGFβ1, and TIMP-1 (% to intact mito-DAMP-treated controls, two-tailed $p$ values indicated, unpaired $t$ test, $n = 6$ of biological replicates derived from two independent experiments performed in triplicates). Data are expressed as means ± SEM. *$P < 0.05$; **$p < 0.01$; and ***$p < 0.001$ compared to vehicle-treated controls (one-way ANOVA, followed by Dunnett's post test). Source data are provided as a Source Data file.

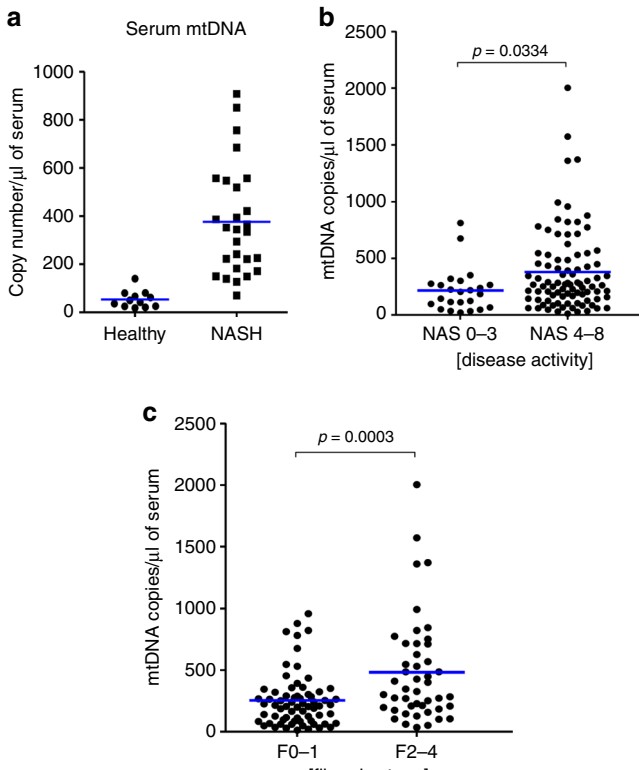

**Fig. 7 Circulating mito-DAMP levels are elevated in patients with NASH in association with histologically significant hepatic fibrosis.** Characteristics of the healthy human subjects and treatment-naive patients with biopsy-proven NAFLD/NASH are summarized in Table 1. **a** Serum levels of mtDNA are markedly elevated in the sera of patients with NAFLD/NASH (cohort I, $n = 27$) compared to healthy subjects ($n = 11$). Increased mtDNA levels in relation to histological NAFLD/NASH disease activity (**b**) and fibrosis (**c**) (cohort II, $n = 114$). mtDNA (D-loop region) was quantified by real-time TaqMan PCR in total DNA isolated from serum as described in "Methods" (dot-plot depict individual values, bar shows group average, two-tailed $p$ value indicated, unpaired $t$ test). Source data are provided as a Source Data file.

Here, we investigated in detail the fibrotic response in inbred strains in response to hepatotoxin TAA-induced liver injury. Similar to previous chronic liver injury studies[2,3,16], BALB/c was most fibrosis-susceptible strain in our system, C57Bl/6 developed intermediate degree of fibrosis, and FVB mice were fibrosis-resistant (Supplementary Fig. 1). Careful analysis revealed that such strain-specific fibrotic responses can be detected using objective quantitative biochemical and histological methods 5–8 days after merely one sub-lethal TAA insult (Fig. 1). Strain differences were not unique to TAA toxicity, because another hepatotoxin, CCl4, resulted in essentially identical phenotype (Supplementary Fig. 2). While results obtained in this experimental system, representing extreme fibrosis phenotypes in FVB and BALB/c mice, cannot be directly extrapolated to human population, we felt such robust and malleable model may allow elucidation of fundamental cellular and molecular mechanisms that would otherwise be challenging to pinpoint in genetically and environmentally complex human disease condition with frequent co-morbidities.

Indeed, longitudinal analysis of responses to acute liver injury in these strains has been revealing. TAA caused death in 40–50% of pericentral hepatocytes, with very similar cumulative liver injury in both resistant and susceptibly strains as assessed via area under the curve (AUC) of serum ALT levels (Fig. 2). Likewise,

both strains were able to mount robust, but nearly identical early transcriptional activation of fibrosis-related genes at day 3. However, in late recovery stages (between days 5 and 8), strain differences became apparent: key pro-fibrogenic mRNAs encoding TGFβ1, collagen α1(I), TIMP-1, and MMP-2 normalized in FVB, but persisted at elevated levels in BALB/c (Fig. 2c). This was preceded by remarkable delay in macrophage-mediated clearance of necrotic hepatocytes due to impaired macrophage efferocytosis function in susceptible BALB/c mice (Fig. 3), release of mito-DAMPs (Fig. 5a), and subsequent activation of HSCs adjacent to the areas of incomplete efferocytosis (Figs. 2 and 3). Thus, it is not the magnitude of early fibrogenic response, but the rapidity and completeness of repair that is a central characteristic of scarless recovery, with efficient phagocytic clearance of dead hepatocytes being the major determinant in our system.

Although phagocytic function of macrophages was not systematically studied in human liver disease, isolated reports have found evidence of impaired macrophage phagocytosis in cirrhotic patients with ascites[17,18]. Using an experimental model of surgical recovery from bile duct ligation-induced fibrosis in rats, we have previously observed that phagocytic macrophages promote (biliary) fibrosis reversal via the release of pro-fibrolytic MMPs upon engulfment of apoptotic cholangiocytes[19]. Earlier in vitro studies reported that HSC line LX-1 is able to engulf apoptotic bodies of hepatocytes, which induced pro-fibrogenic activation in vitro via a nicotinamide adenine dinucleotide phosphate oxidase[20] and phosphoinositide 3-kinase/p38 mitogen-activated protein kinase-dependent manner[21]. Many cell types are indeed capable of slowly engulfing the injured/dead cell in their close proximity[22]. However, while apoptotic cell engulfment by stellate cells was definitively demonstrated to occur in vivo[20], macrophages are dramatically more efficient at efferocytosis than any other stromal cells. When considering their unique ability to actively sense and find dead cells, it is likely that in the physiological in vivo setting, resident and infiltrating monocyte/macrophages easily outcompete HSCs in clearing dead hepatocytes.

The central role of macrophages in regulating liver fibrosis is now well recognized[11,23]. It is also complex and often paradoxical, since genetic depletion of the same CD11b(+) macrophage subset had opposite effects on fibrosis when performed during chronic liver injury or the recovery phase[15]. Recent liver fibrosis research focused mainly on infiltrating monocyte/macrophages recruited to the liver from bone marrow, with some of specific subsets identified as functionally promoting fibrosis progression (CD11b(+)F4/80(+)Gr-1(+))[24] or reversal (CD11b (hi)F4/80(int)Ly-6C(lo))[25]. Interestingly, pro-resolution infiltrating CD11b(hi)F4/80(int)Ly-6C(lo) MΦ subset demonstrate a post-phagocytic phenotype, which was further enhanced by the administration of non-loaded liposomes in vivo[25]. Although the phagocytic function was not specifically assessed in these studies, it can be speculated that divergent regulation of phagocytic ability monocytes/macrophages during inflammation and repair may explain opposite MΦ roles[15]. Here, via several experimental approaches we demonstrate that efficient phagocytic function of resident macrophages and Gr-1+ myeloid is critical to fibrosis resistance in FVB mice, while defective efferocytosis macrophage function in BALB/c (or when phagocytes are depleted in FVB) leads to persistence of dead hepatocytes and exaggerated fibrotic responses (Figs. 2–4). Remarkably, fibrosis-resistant FVB strain that clears dead hepatocytes efficiently can be rendered equally fibrosis-susceptible as BALB/c via selective depletion of resident and infiltrating macrophages based on their phagocytic ability using either CLOs, or by depletion of an infiltrating myeloid cells using anti-Gr-1 antibody (Fig. 4). Surprisingly, and contrary to prior report on genetic CD11b+ cell deletion in CCl4-induced liver fibrosis in mice[15], antibody depletion of CD11b(+) cells had

no direct effect on dead hepatocyte clearance and ensuing fibrotic response in our system (Fig. 4e, Supplementary Fig. 6). This may suggest that pro-fibrogenic role of infiltrating CD11b+ monocytes pool might be indirect and determined by their well-established pro-inflammatory function[26]. Because efferocytosis is highly redundant both in terms of capable cell types and molecular machinery of engulfing dead cells[27], another possible explanation is that antibody recognizing more broadly expressed Gr-1 (Ly-6G/Ly-6C, compared to CD11b) massively depleted phagocytes, which could not be fully compensated by resident macrophages or Gr-1(−) subsets. Alternative possibility, although less likely, is the existence of minor Gr(+)CD11b(−)Ly-6G(−) subset with critical phagocytic activity; detailed follow-up study is warranted to address this question. Importantly, repeated CLO-mediated phagocytes depletion performed concurrently with chronic TAA administration resulted in significant fibrosis in (normally resistant) FVB mice, suggesting that the same mechanism applies in the clinically relevant settings of chronic liver disease (Supplementary Fig. 5).

The striking association of fibrotic responses with delayed clearance and thus persistence of dead hepatocytes (Figs. 1 and 2), as well as HSC accumulation and collagen deposition within these areas of "failed" efferocytosis (Fig. 3), led us to hypothesize that intracellular contents leaking from hepatocytes are directly responsible for triggering fibrotic response. Indeed, the role of DAMPs released from damaged hepatocytes in initiating or amplifying an inflammatory response, analogous to pathogen-associated molecular patterns of infectious pathogens in sterile liver injury, is now well recognized[28]. Mitochondria-derived DAMPs (mito-DAMPs), which are particularly immunogenic due to structural similarities to bacteria, have attracted attention recently[8]. In an extreme example such as massive trauma, mito-DAMPs, identified as mtDNA and formyl peptides, are acutely released from damaged tissues and cause a sepsis-like systemic inflammatory response syndrome by activating polymorphonuclear cells via TLR9 and formyl peptide receptor-1[29]. In the liver, where hepatocytes are extremely rich in the mitochondria, mtDNA was shown to be released into circulation and to promote inflammation in NASH through TLR9 ligation[30] and exacerbate ischemia–reperfusion injury[31]. However, whether mito-DAMPs can directly trigger fibrotic response was not investigated until now. In our system, mito-DAMPs release into circulation (measured via serum mtDNA levels) was evident on day 2 post injury only in susceptible BALB/c strain (Fig. 5a), coinciding with failure to upregulate several key phagocytosis molecules, diminished efferocytosis (Fig. 3), and delayed dead hepatocyte removal (Fig. 2). In order to test whether mito-DAMPs are indeed the culprit, we purified crude soluble mito-DAMPs from liver mitochondria and administered mito-DAMPs into resistant FVB mice during recovery from TAA to mimic prolonged exposure to dead cell content/mito-DAMPs in BALB/c strain. This resulted in remarkable activation of HSCs, up-regulation of pro-fibrotic transcripts, and fibrotic scar formation by day 8 of recovery, effectively abrogating fibrosis resistance in FVB mice (Fig. 5). Importantly, we were able to confirm robust HSC activation in response to a single administration of exogenous mito-DAMPs in a mechanistically different model of early-stage MCD-induced steatohepatitis in C57Bl/6 mice (Fig. 5g). Because mito-DAMPs were able to dose-dependently activate freshly isolated HSCs in vitro (Fig. 6), this appears to be a direct, primary effect on fibrogenic effector cells (as opposed to secondary, e.g. due to immune cells activation). Importantly, we identified mtDNA as a major active moiety within mito-DAMPs, accounting for about half of their fibrogenic activity (Fig. 6d–h). Future studies should elucidate the molecular sensors mediating such effects of mtDNA, as well as the contribution and potential synergies of other

components of the mitochondria. Resident liver macrophages regulate fibrotic response and clear dead hepatocytes (Fig. 4), and are presumably exposed to high levels of mito-DAMPs in liver disease. However, our in vitro data show that unlike HSCs, primary KCs do not respond to equivalent doses of mito-DAMPs in classical activation assays; moreover, their activation by IFNγ or LPS was not modulated in the presence of mito-DAMPs (Supplementary Fig. 8). This is potentially due to naturally high exposure (and relative tolerance) of resident liver macrophages to gut-derived bacterial products, which bear structural resemblance to mito-DAMPs. To our knowledge, this is the first demonstration that mito-DAMPs can directly activate myofibroblastic cells and trigger pro-fibrogenic response independently their well-characterized pro-inflammatory action.

Importantly, we demonstrate that in human chronic liver disease, mito-DAMPs can escape from local hepatic microenvironment into circulation. mtDNA, a biologically active component of mito-DAMPs (Fig. 6d–h), is detected at very low levels in healthy subjects, but is dramatically elevated in the serum of human NAFLD/NASH patients (Fig. 7a), supporting the relevance of proposed mito-DAMPs pathway in driving human hepatic fibrosis. Furthermore, mtDNA levels are further increased in patients with active NASH (NAS 4–8 versus NAS 0–3, $p = 0.0334$) and demonstrate stronger association with significant (F2–4) versus minimal/no fibrosis (F0–1, $p = 0.0003$) on liver biopsy (Fig. 7b, c). These findings have immediate translational implications as novel disease biomarker; the diagnostic potential of circulating mtDNA species to identify patients at risk of rapid progression to cirrhosis should be also evaluated in prospective sequential liver biopsies studies.

Our results indicate that efferocytosis is a fundamental mechanism guarding the liver from excessive fibrotic response to toxic insults by limiting exposure to mito-DAMPs, with intriguing, previously unrecognized therapeutic implications. However, clearance of dying cells is a highly complex and orchestrated process involving interactions between multiple ligands on dying cells, bridging molecules, and receptors on phagocytes with a high degree of redundancy[32]. Comprehensive profiling of 84 known phagocytosis genes 48 h after liver injury (coinciding with systemic release of mito-DAMPs, Fig. 5a) revealed inefficient activation of specific set of genes governing multiple steps in the phagocytic pathway (Fig. 3c, d). Some of these genes have been previously implicated in liver fibrosis (Serpine1[33], Tgm2[34]), while role of others (Cd14, Marco, Csf1) have not been investigated to date. Identification of such candidate genes is critical to begin elucidation of underlying molecular phagocytosis pathways both in mouse and human species. Demonstration of the crucial role of efferocytosis, identification of effector cell subset(s), and phagocytic genes (Figs. 2–4) in our report pave the way for such studies.

Conceptually, results of our study support the idea that excessive fibrotic response is a direct result of a "failed" resolution of sterile inflammation[35], with impaired phagocytosis of injured hepatocytes enabling exposure of quiescent stellate cells to fibrogenic mito-DAMPs. Robust phagocytic genes activation, followed by rapid efferocytosis of injured hepatocytes and MΦ departure (as observed in resistant FVB mouse) terminate fibrotic response early and result in scarless repair. Conversely, activation of multiple phagocytic genes is impaired in susceptible BALB/c strain (or when phagocytes are depleted), which result in inefficient dead hepatocyte clearance, fibrogenic mito-DAMPs leak into extracellular space, which trigger HSC activation and scar formation (Supplementary Fig. 9). It is possible that alternative mechanisms, such as epigenetic, microbiota, intestinal epithelial integrity, or portal endotoxin levels may also contribute to susceptibility to liver fibrosis in mice and humans. However, the extent to which fibrosis resistance is compromised either by

phagocyte depletion (Fig. 4) or circumventing phagocytic clearance by exogenous mito-DAMPs administration (Fig. 5), together with increased mito-DAMPs levels in human fibrotic NASH (Fig. 7), strongly suggests that mito-DAMPs released from hepatocyte escaping phagocytic clearance play a major, previously unrecognized role in liver fibrosis.

Together, these results provide a new mechanistic insight into the pathogenesis of liver fibrosis, and offer a conceptual framework for future studies into molecular mechanisms mediating susceptibility to liver fibrosis. Based on our results, circulating levels of mito-DAMPs species can be leveraged as biomarkers of progressive disease in human NAFLD. Finally, therapeutic targeting of mito-DAMPs release or modulation of phagocytosis may serve as a promising approach to treat liver fibrosis.

## Methods

**Animal experiments**. Male 6–7-week-old FVB/NJ (stock #001800), C57BL/6J (#000644) inbred mice were purchased from Jackson Labs (Bar Harbor, ME); male 6–7-week-old BALB/c (stock #028) mice were obtained from Charles River (Wilmington, MA). All mice were acclimatized for 1 week before experiments and housed on a 12 h dark and light cycle at 18–22 °C with 40–60% humidity, and fed a standard rodent chow and tap water *ad libitum*. Animal experiments had been reviewed and approved by the Beth Israel Deaconess Medical Center's Institutional Animal Care and Use Committee (IACUC) (protocols #158-2008, #004-2012, and #010-2015).

*Chronic model of TAA-induced liver fibrosis* was induced in 7–8-week-old male mice by repetitive injections of TAA. Briefly, fibrosis was induced by chronic i.p. TAA injection for 6 weeks (first dose—100 mg/kg, all subsequent doses 200 mg/kg three times a week). Mice were sacrificed always 3 days after the last TAA administration, unless specified otherwise.

*Model of recovery from a single sub-lethal dose of TAA* was established in pilot experiments in each strain. Half of the dose that caused ~50% mortality in FVB/NJ (100 mg/kg of TAA in PBS) was chosen and injected i.p. once in all subsequent experiments and recovering mice sacrificed 1–8 days post injury. In selected experiments, sub-lethal liver injury was induced by a single injection of another hepatotoxin, CCl4 (0.8 mg/kg, dissolved in mineral oil via oral gavage).

*Steatohepatitis model* was induced by feeding MCD for 2 weeks in male 8-week-old C57Bl6/J mice (Jackson Labs), which results in early steatohepatitis without fibrosis or significant HSC activation[36].

**Macrophage and myeloid cell subsets depletion in vivo**. Selective macrophage depletion was achieved with a single i.p. injection of CLOs (10 µl/g of body weight, https://clodronateliposomes.com/). This method of macrophage depletion was chosen based on its unique ability to target macrophages based on their phagocytic activity[37], as opposed to their genetic "lineage" depletion[15]. Efficiency of liver macrophage depletion was confirmed by almost complete disappearance of F4/80+ hepatic macrophages as assessed by immunofluorescence (Supplementary Fig. 4). CLO-mediated depletion of phagocytic macrophages was performed in two regimens: to deplete resident hepatic macrophages (KCs), CLO was administered 24 h prior to TAA injection; depletion of both resident and infiltrating monocytes/macrophages was achieved by CLO injection 24 h after TAA injury. In selected experiments, chronic macrophage depletion was performed by weekly injection of CLO (10 µl/g of body weight) during progression of chronic TAA-induced fibrosis (induced as described above). Liposomes loaded with PBS served as controls, unless otherwise stated. Monoclonal antibody-mediated depletions of myeloid cell subsets were performed according to published protocols using Gr-1 mAB (RB6-8C5, 200 µg/mouse), CD11b mAB (M1/70, 200 µg/mouse)[38], Ly-6G mAB (1A8, 500 µg/mouse)[39], or isotype control (LTF-2 IgGb2b, 200 µg/mouse) (all from BioXcell, Inc.) administered intraperitoneally at 1, 3, and 5 days post TAA in FVB mice.

**In vivo phagocytosis assay**. Fluorescently labeled apoptotic thymocytes were prepared as described[40]. Briefly, thymocytes were isolated by mechanical dissociation of thymi from 3- to 4-week-old, male recipient strain-matched mice, labeled with Cell Tracker™ Green CMFDA Dye (Thermo Fisher, #C7025) according to the manufacturer's manual, washed, and apoptosis induced by incubation with 1 µM dexamethasone for 16 h at 37 °C and 5% CO2. Apoptosis was confirmed by annexin V binding and morphological changes (nuclear fragmentation/condensation, membrane blebbing) and was routinely observed in ~50% cells (<30% viable cells as assessed by propidium iodide exclusion). A total of $2 \times 10^6$ apoptotic thymocytes were injected into the tail vein of strain-matched recipient mouse (either healthy controls or 48 h post-TAA-induced liver injury), which were euthanized 1 h later. Engulfment of apoptotic thymocytes (green) was analyzed in snap-frozen, acetone-fixed liver sections immunofluorescence stained for F4/80 or Gr-1 (red)[41]. Ten randomly chosen high-power fields at ×200 magnification per animal were assessed.

**Phagocytosis gene array**. Total liver RNA was extracted using the RNEasy mini-columns (Qiagen). Complementary DNA was prepared from 0.5 mg of RNA using the RT2 first-strand kit and quantitative PCR for 84 phagocytosis-related genes was then performed using the RT2 Profiler mouse phagocytosis PCR array (PAMM-173ZA-24, Qiagen) on ABI7000 qPCR instrument according to the manufacturer's protocol. The data obtained were exported to the Qiagen's GeneGlobe Data Analysis Center software-based tool where it was normalized to β-2 microglobulin (*β2MG*) gene and analyzed using the ΔΔCt method according to the manufacturer's instructions.

**Mito-DAMPs preparation, characterization, and administration**. Liver mitochondria were isolated and purified from 6- to 8-week-old FVB mice according to a standard method of differential centrifugation in a sucrose gradient as described[42]. To prepare soluble mito-DAMPs, mitochondrial pellets were re-suspended in 20 mM PBS (274 mM NaCl, 20 mM PO4³⁻, 5.4 mM KCl; pH 7.4) and incubated on ice for 15 min, followed by 10 cycles of freezing–thawing (freezing in liquid nitrogen for 30 s/thawing at 20 °C for 1 min). Supernatant (mito-DAMPs) was collected after centrifugation at $15000 \times g$ for 10 min and characterized by mtDNA content, purity and integrity, and total protein content. Purity and integrity of mtDNA was assessed by the percent of total DNA found in 16 Kbp band via automated microgel electrophoresis (Agilent 4200 TapeStation). Potential contamination with nuclear DNA was controlled by TaqMan RT-PCR quantification of genomic (nuclear) reference region within *β2MG* in parallel with mitochondrial 12S and 16S regions, and was routinely found to be below 1%. Protein content was determined by the Bradford assay (Abcam, Cambridge, MA). In all experiments, mito-DAMP preparations were standardized by intact mtDNA concentration as determined by automated DNA microgel densitometry on Agilent 2200 TapeStation system. Mito-DAMP preparations with mtDNA integrity of >95% were diluted at 0.1 µg mtDNA/µl concentration and stored at −80 °C until the experiment. In vivo, mito-DAMPs (at the dose of 9.5 µg mtDNA/mouse, which corresponds to 10% of the liver) were injected intraperitoneally twice, on the second and fourth day of recovery after acute TAA injury in mice. Dosage and i.p. route of administration was chosen based on mito-DAMP's stability and favorable systemic exposure mimicking physiologic mito-DAMP levels (Supplementary Fig. 7) In vitro, freshly isolated HSCs were treated on day 2 with mito-DAMPs at increasing doses (corresponding to mtDNA content of 0.25, 0.5, and 0.75 µg/ml) and their activation state analyzed 24 h later. mtDNA depletion was performed by pre-incubation of mito-DAMP preparations with RQ1 RNase-free DNase (1 U DNase/1 µg mtDNA, Promega, cat. # M610A) for 30 min at 37 °C prior to addition to cells.

**Primary non-parenchymal liver cell isolation**. Primary HSCs were isolated from fibrotic *Mdr2−/−* mice and cultured on plastic[43]. Briefly, after in situ perfusion of the liver with pronase (Roche, Indianapolis, IN), followed by collagenase (Roche), dispersed cell suspensions were layered on a discontinuous density gradient of 29% Nycodenz (Sigma). HSCs were collected from the gradient interface, and cell viability was verified by phase-contrast microscopy as well as the ability to exclude trypan blue. The viability of all cell cultures used for the studies was >95%. Primary KCs were isolated similarly to HSCs, except for gradient centrifugation, which was performed as described[44] with modifications. Briefly, non-parenchymal cell pellets were re-suspended and centrifuged on a density cushion of Percoll (25 and 50%). Cell fraction enriched in KCs located between 25% and 50% Percoll was collected and seeded in 12-well plates at a density of $1 \times 10^6$/well. After 20 min incubation, non-adherent cells were removed. Cell purity was assessed by latex bead uptake. Cell preparations with purity and viability over 95% were used for further studies. Activation assays were performed 24 h after isolation in the presence of 10 µg/ml LPS (R&D), 100 U/ml IFNγ (R&D), or mito-DAMPs (corresponding to 2.5–7.5 µg of mtDNA/ml) for 24 hs, using IL-1, TNF-α (ELISA, R&D) and NO secretion (colorimetric assay, Thermo Fisher Scientific) into cell supernatant as read-outs.

**Hepatic hydroxyproline determination**. Hepatic collagen content was determined as relative hydroxyproline (µg/g liver) in 250–300 mg liver samples from two different lobes (representing >10% of whole liver) after hydrolysis in 6 N HCl for 16 h at 110 °C using colorimetric Cholamine-T/p-dimethylaminobenzaldehyde method[45]. Total hydroxyproline (mg/whole liver) was calculated based on individual liver weights and the corresponding relative hydroxyproline content[19,46].

**Quantitative real-time RT-PCR**. Liver tissue (250–300 mg) from two lobes was homogenized and total RNA was extracted using RNAPure (PeqLab, Erlangen, Germany), and 1 µg of total RNA reverse transcribed[46,47]. Relative transcript levels were quantified by real-time RT-PCR on a LightCycler 1.5 instrument (Roche, Mannheim, Germany) using the TaqMan methodology. TaqMan probes (dual-labeled with 5′-FAM and 3′-TAMRA) and primers (summarized in Supplementary Table 1) were designed using the commercial Primer Express software version 3.0 (Perkin-Elmer, Wellesley, USA), synthesized at Eurofins (Louisville, KY), and validated previously[46,47]. The housekeeping gene *β2MG* was amplified in parallel reactions for normalization.

**Immunohistochemistry and immunofluorescence**. Indirect immunofluorescence was performed in frozen acetone-fixed sections, and immunohistochemistry for α-

SMA on formalin-fixed paraffin-embedded liver sections were performed according to routine protocols[48]. All primary antibodies used are summarized in Supplementary Table 2. Hematoxylin–eosin and Sirius Red staining was performed using standardized clinical pathology protocols at Histology Core (BIDMC).

**Serum biochemistry**. Serum levels of ALT were measured using Catalyst Dx® Chemistry Analyzer (IDEXX Laboratories, Inc., Westbrook, ME) according to the manufacturer's recommendations.

**MTT proliferation assay**. Cell proliferation was measured by a MTT assay (ATCC). Cells were plated at a density of $5 \times 10^3$ cells per well in 96-well culture plates. After treatment, MTT solution was added to the culture medium (0.5 mmol/l) and incubated for 2 h at 37 °C with 5% $CO_2$. Detergent solution was then added to solubilize formazan crystals. Optical density was determined at 540 nm using a Benchmark Plus microplate reader (Bio-Rad, Hercules, USA).

**Human samples**. De-identified serum samples from healthy subjects ($n = 12$) were procured from human samples repository (Innovative Research, Novi, MI). De-identified serum samples from adult patients (age ≥ 18 years) with NAFLD were obtained from a prospective NAFLD patient registry and BioBank at Beth Israel Deaconess Medical Center (Boston, MA). All patients had biopsy-proven NAFLD without other chronic liver diseases or significant alcohol consumption (patients with >20 g alcohol daily were excluded from the registry). The study was reviewed and approved by the Institutional Review Board of Beth Israel Deaconess Medical Center and informed consent was obtained from all study participants. Serum mtDNA levels were analyzed in total DNA isolated from 25 to 50 μl serum (Gentra Puregene kit, Qiagen) by quantitative TaqMan PCR amplification of D-loop region within mtDNA genome in pilot (cohort I, $n = 27$) and validation cohort (cohort II, $n = 114$) in relation to histological disease activity and fibrosis. External intact mtDNA standards isolated from human HepG2 hepatoma cells' mitochondria were used to calculate target's copy number. All biopsies were scored according to Kleiner and Brunt et al.[49] and steatohepatitis activity was defined by NAS score as no/minimal (NAS 0–3) or active NASH (NAS 4–8), whereas fibrosis was defined as no/minimal (F0–1) or significant (F2–4). Demographic and clinical characteristics are summarized in Table 1.

**Statistical analyses**. Data are expressed as means ± SEM, and statistical analyses were performed using Microsoft Excel Office 365 and GraphPad Prism version 8.4.0 (GraphPad Software, San Diego, CA). Student's $t$ test was performed for two-group comparisons. Multiple comparisons were performed by one-way ANOVA (analysis of variance, followed by Dunnett's or Tukey's post test) and S. Two-tailed $p$ values < 0.05 were considered significant and are reported in the graphs.

**Reporting summary**. Further information on research design is available in the Nature Research Reporting Summary linked to this article.

## Data availability

The source data underlying Figs. 1b, c, 2c, 3b–d, 4c, d, 5a, d–g, 6b, c, f–h, 7a–d, Supplementary Figs. 1b, 2b, 3a, 5b, 6b, 7b–e, 8a, b are provided as a Source Data file. All other data are available from the corresponding author upon reasonable request.

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

## Acknowledgements

We are grateful to Konstantin Khrapko (HMS/Northeastern University) for sharing valuable insights and the expertise in mitochondrial genome and structure, to Henryk Koziel (BIDMC/HMS) for the help in designing in vitro macrophage experiments, and to Barbara Wegiel (BIDMC/HMS) for critical discussions of myeloid cell subsets depletion. We also thank Rohan Akhouri, Thomas Konturek, Michael Tandetnik, Michael Lynch, and Julie Shea (BIDMC) for their expert technical assistance. This work was supported by Irving W. and Charlotte F. Rabb Award, research grant from PSC Partners Seeking a Cure Canada, an institutional pilot grant from Department of Medicine, Beth Israel Deaconess Medical Center to Y.V.P., and a fellowship by National Natural Science Foundation of China (81302131) to P.A.

## Author contributions

P.A. and L.-L.W.—experimental design, analysis, statistical analysis, and drafting of the manuscript; S.Z. D.Y.S., K.A.V., M.M., and K.K.—selected assays and analysis; M.L.—clinical samples collection and analysis, Y.V.P.—study concept, design and supervision, data analysis, drafting, and editing the manuscript.

## Competing interests

The authors declare no competing interests.
