## [Peer Review File · Nature Communications]

Reviewers' comments:

Reviewer #1 (Remarks to the Author):

The study by An et al is very fascinating and original for the second part of the study regarding the role of mito-DAMPs of dying hepatocytes as a triggers of fibrosis.

The results are well presented and performed. However, there are several criticisms.

The manuscript seems a composition of different studies and the link between the different parts is completely missing.

The first part on the susceptibility of the different mouse strains to fibrosis with aim to explain the differences between humans is little hard to believe. Susceptibility is a concept that in humans could be linkend to genetics and epigenetics and all hereditary traits. However, I suggest to the authors a reordering of the text by the speculation that mito-DAMPs could be differents in susceptible strains because of mutations in mitochondrial DNA.

Finally, the part on patients with NASH is limited. Sample size is not sufficient to extrapolate data on fibrosis and the correlations with the histologic pattern of subjects is lacking.

The methods reported in the main text are poor.

Reviewer #2 (Remarks to the Author):

Popov and colleagues used fibrosis-resistant FVB and fibrosis-susceptible Balb/c mice to investigate the molecular mechanism of mito-DAMP-mediated liver injury and fibrosis in TAA-treated mice. The authors demonstrated that FVB and Balb/c mice showed different responses to fibrotic injury. These mice showed different clearance of dead hepatocytes and different patterns of infiltration of liver macrophages. Moreover, they also found the releases of mito-DAMPs were different between these two mouse strains. Mito-DAMP treatment and depletion of liver macrophages and Gr-1 expressing cells resulted in the exacerbation of liver fibrosis in FVB mice. Also, they showed mito-DAMPs promoted hepatic stellate cell activation in vitro. Notably, circulating mito-DNAs were increased in NASH patients. The clinical relevance of this study is high. The manuscript is well-written and the data presentation is also well-organized. However, why these two mouse strains show different mito-DAMP release in hepatocytes and different recruitment styles of macrophages and myeloid cells are still unclear.

Major comments:

1. The authors showed different mito-DNA levels between FVB and Balb/c strains. How did the authors determine these mito-DNAs were derived from hepatocytes? This result suggests the susceptibility of liver injury between FVB and Balb/c strains to TAA is different. What is the mechanism? Alternatively, are the mitochondrial DNA copy numbers different between strains?
2. The authors extracted whole mito-DAMPs. How did the authors know the responsible component of mito-DAMPs is mito-DNA, but not other mito components? The contribution of TLR9 or the other sensors for mitochondrial proteins should be examined.
3. The recruitment patterns of macrophages are different between strains. Is this explained by different amounts of mito-DNA release? Or do other cytokine or chemokine productions contribute?

Point by point response to reviewer's comments:

We are grateful to both reviewers for insightful critical comments and enthusiasm for our study. We performed extensive additional experiments to address the points raised, which we believe clarified important aspects of studied pathways, substantially strengthened our initial conclusions and led to an overall improvement of our work. Please find below our detailed responses to specific reviewers comments, as well as relevant changes made to figures and the text of the manuscript.

Reviewer #1:

General comment: *“The study by An et al is very fascinating and original for the second part of the study regarding the role of mito-DAMPs of dying hepatocytes as a triggers of fibrosis. The results are well presented and performed. However, there are several criticisms. The manuscript seems a composition of different studies and the link between the different parts is completely missing.”*

A: We appreciate reviewer enthusiasm for our manuscript and regret the impression that different aspects of our study are not well connected. The order in which we report our results reflects the actual sequence of experiments that lead to the comprehensive characterization of the pathway and we feel it is best reported in this way. During revisions, we specifically aimed at refining phagocytosis part, which we postulate as a main point-of-control for extracellular/systemic mito-DAMPs release, and ensuring the smoother transition between different parts of the study, which we hope together alleviates reviewer's concern and communicates our findings and concept more clearly.

Q1: *“The first part on the susceptibility of the different mouse strains to fibrosis with aim to explain the differences between humans is little hard to believe. Susceptibility is a concept that in humans could be linked to genetics and epigenetics and all hereditary traits. However, I suggest to the authors a reordering of the text by the speculation that mito-DAMPs could be different in susceptible strains because of mutations in mitochondrial DNA”*

A1: The point of reviewer on a complex nature of susceptibility to fibrosis, and possible role of epigenetic and environmental factors (in addition to influence of genetic background we addressed in this study) is well taken. We agree that the susceptible/resistant inbred mouse strains system we utilized is somewhat simplistic to directly extrapolate to heterogeneous human population (species differences also cannot be ruled out). However, we are also convinced that such robust and malleable models allow to elucidate fundamental cellular and molecular mechanisms that would otherwise be challenging to pinpoint in genetically and environmentally diverse and complex human disease condition (which is further complicated by frequent co-morbidities). The novel pathway we described in our study will undoubtedly require further investigation into precise molecular mechanisms and validation in humans, which is beyond the scope of present study but is actively pursued by our group. We now refined our Introduction and Discussion sections accordingly (changes underlined).

In regards to second part of reviewer's comment, functional significance of mutational load in mtDNA is an active area of basic research in aging and disease and it may, at least theoretically, play a role in susceptibility to fibrosis. However, due to the fact that our study focused on extracellular release of mitochondrial DAMPs products (including mtDNA) regardless of their mutational load or mitochondrial function per se, we did not discuss this aspect in our manuscript. In fact, prior studies found very little mutational variability in mtDNA genes in studied laboratory mouse strains despite high variability in fibrosis susceptibility, which all derived from a single mtDNA ancestor less than 100 years ago via inbreeding (see for example *Zheng et al. Mitochondrion. 2014 Jul;17:126-31*). Thus, such sequence variability (or potential variance in mitochondrial to nuclear genomes ratio, please refer to our reply to

comment 1 by reviewer 2) is unlikely to explain profound differences we observe in terms of susceptibility to fibrosis.

Instead, we demonstrate that susceptibility to liver fibrosis (Fig. 1) in our system is due to significant strain differences in mounting phagocytosis-related gene expression and efferocytosis function in response to liver injury, resulting in delayed clearance of dead hepatocytes (Fig. 2, 3, 4) and extracellular leak of mito-DAMPs which directly trigger fibrogenic activation of stellate cells (Fig. 5, 6) in fibrosis-susceptible mice. We now performed comprehensive additional experiments, which include direct demonstration of impaired efferocytosis post-injury in fibrosis-susceptible BALB/c mice using in vivo phagocytosis assay (new Fig. 3BC, new Tab. S3), as well as identification of putatively responsible phagocytosis gene set via gene profiling data (new Fig. 3DE, new Tables S4 and S5) with TaqMan cross-validation (new Table S6). These new data confirm and strengthen our original conclusions. Such profound strain differences in mounting phagocytic response are most likely due to specific genomic differences that remain to be mapped. We re-organized and edited the text of abstract and all sections of main text to incorporate new data and more clearly and logically connect all aspects of our study (changes are underlined).

Q2: *“Finally, the part on patients with NASH is limited. Sample size is not sufficient to extrapolate data on fibrosis and the correlations with the histologic pattern of subjects is lacking”*

A2: We now analyzed an additional cohort of 114 NAFLD patients with results included as new Fig. 7, along with previous results in pilot NAFLD cohort of 27 patients and 12 healthy controls. This new set of data clearly demonstrate a (weaker, $p=0.0334$) association of mtDNA levels with active NASH (NAS 4-8) and (stronger, $p=0.0003$) association with significant fibrosis on liver biopsy (defined as $\geq F2$); lending further support to our original conclusions and the relevance of newly identified pathway to human disease. These findings have an obvious diagnostic implications which will be the focus of follow-up studies; mito-DAMPs pathobiology we characterized in this study suggests that circulating mtDNA may best reflect the dynamic rate of fibrosis progression (fibrogenesis *activity*) rather than cumulative fibrogenesis *outcome* (fibrosis stage). We recognize that ideal system to evaluate the diagnostic potential of circulating mtDNA species would be in relation to its ability to predict future histological fibrosis progression rate, calculated based on fibrosis staging in sequential liver biopsies. We initiated such prospective study and discuss new results presented in new Fig. 7 and its implications in Discussion section. We thank reviewer for this comment which led to significantly strengthened translational impact and rigor of our study.

Q3: *“The methods reported in the main text are poor”*

A3: Due to space constraints, we initially included only methods that are non-routine and most essential to understanding of this study (e.g. phagocytosis- or mitoDAMPs-related), with complete description of standard laboratory techniques listed in supplementary material. We have now incorporated full methods in the main manuscript, as requested.

Reviewer #2 (Remarks to the Author):

General comment: *“Popov and colleagues used fibrosis-resistant FVB and fibrosis-susceptible Balb/c mice to investigate the molecular mechanism of mito-DAMP-mediated liver injury and fibrosis in TAA-treated mice. The authors demonstrated that FVB and Balb/c mice showed different responses to fibrotic injury. These mice showed different clearance of dead hepatocytes and different patterns of infiltration of liver macrophages. Moreover, they also found the releases of mito-DAMPs were different between these two mouse strains. Mito-DAMP treatment and depletion of liver macrophages and Gr-1 expressing cells resulted in the exacerbation of liver fibrosis in FVB mice. Also, they showed mito-*

DAMPs promoted hepatic stellate cell activation in vitro. Notably, circulating mito-DNAs were increased in NASH patients. The clinical relevance of this study is high. The manuscript is well-written and the data presentation is also well-organized. However, why these two mouse strains show different mito-DAMP release in hepatocytes and different recruitment styles of macrophages and myeloid cells are still unclear. “

A: We appreciate this reviewer's enthusiasm for our manuscript and regret the lack of clarity on mechanistic explanation of observed strain differences in mito-DAMPs release. In particular, as noted also by reviewer 1, critical role of phagocytosis of dead hepatocyte in limiting mito-DAMPs release was suggested based on descriptive evidence but was not directly demonstrated in functional experiments. During revisions, we especially aimed at addressing the role of phagocytosis in greater depth and detail, which we postulate as a main point-of-control for extracellular/systemic mito-DAMPs release, and ensuring the smoother transition between different parts of the study, which we hope together alleviates reviewer's concern and communicates our findings more clearly.

Q1: *“Major comments: The authors showed different mito-DNA levels between FVB and Balb/c strains. How did the authors determine these mito-DNAs were derived from hepatocytes? This result suggests the susceptibility of liver injury between FVB and Balb/c strains to TAA is different. What is the mechanism? Alternatively, are the mitochondrial DNA copy numbers different between strains?”*

A: Although the cellular source of mtDNA in the serum cannot be experimentally and unequivocally discerned (due to virtually identical mitochondrial genome sequence in liver compared to other cells in the body), there is extremely low probability it originated from cells other than hepatocytes. Thioacetamide (TAA) is a potent hepatotoxin, which generates highly reactive radical upon its metabolism by a hepatocyte's cytochrome P450 system (see for example, PMID: 22867114). This generates massive cell death specifically in hepatocytes (about 50% across strains in our system) which is, in all likelihood, a main if not the only source of mtDNA that escapes into the circulation in susceptible BALBc strain.

Regarding the second part of reviewers comment, we would like to respectfully point out that susceptibility to TAA-induced injury was carefully assessed both histologically and by performing quantitative area under curve analysis of serum ALT and did not differ between strains (see Fig. 1C and Fig. 2A). We postulate in our study that the main reason for mito-DAMPs escaping into circulation is impaired efferocytosis of injured/dead hepatocytes, which persist for several days longer in BALB/c compared to FVB strain (Fig. 2). We now performed comprehensive additional experiments, which include a direct demonstration of impaired function of efferocytosis post-injury in fibrosis-susceptible BALB/c mice using in vivo phagocytosis assay (new Fig. 3BC, new Tab. S3), as well as identification of putatively responsible phagocytosis gene set via gene profiling data (new Fig. 3DE, new Tables S4 and S5) with TaqMan cross-validation (new Tab. S6).

Finally, as requested we have performed mtDNA copy number assessment relative to nuclear DNA, which did not differ between the strains (reported in new supplemental Fig. S7DE). Text was amended accordingly with changes tracked.

Q2: *“The authors extracted whole mito-DAMPs. How did the authors know the responsible component of mito-DAMPs is mito-DNA, but not other mito components? The contribution of TLR9 or the other sensors for mitochondrial proteins should be examined.”*

A. These are excellent and logical questions, which we already actively pursue in a follow up studies focusing on individual mito-DAMPs components and their potential synergy, as well as molecular sensors and transduction pathways involved.

In original manuscript, we did not suggest that mtDNA is solely responsible component of mito-DAMPs, although it does appear to play a major role. We now performed new in vitro mito-DAMPs experiments (with or without DNase 1 pre-treatment) which clearly and directly demonstrate mtDNA is

a major active component of mito-DAMPs, accounting for about 50% fibrogenic activity of mito-DAMPs in vitro (as determined by stellate cell proliferation assay, see new Fig. 5D-H). We are also actively investigating molecular sensing pathways, which appears to be different from previously implicated mechanism (e.g. we found that TLR9 is dispensable for fibrogenic effects on stellate cells of either mito-DAPMs or purified mtDNA both in vitro and in vivo). However, we feel that this extensive body of work in progress will be best reported as a separate follow-up study due to space/time constraints.

Q3: The recruitment patterns of macrophages are different between strains. Is this explained by different amounts of mito-DNA release? Or do other cytokine or chemokine productions contribute?

A. We apologize for lack of clarity in the text of original manuscript on this specific aspect. We did not suggest that recruitment of macrophages/myeloid cells (that occurs very rapidly in early stages, e.g. within first 24h post-injury) is different; instead, the changes in inflammatory infiltrate we observed become apparent in rather late, resolution stages (between day 5 and 8 post-injury). In order to clarify this further, we performed refined analysis of early changes (0, 12 and 24 hours post-injury) for several macrophage/monocyte markers expression (*CD45*, *F4/80*, *Clec4F*, *CD68*, *CD11b*, *CD11c*) which were indeed not substantially different between strains (new Fig. S3A). The key reason behind accumulation of macrophages in susceptible BALB/c strain at late recovery stages appears to be functional (impaired phagocytosis and inability to efficiently clear hepatocyte debris in BALBc), as we directly demonstrate in new experiments (new Fig. 3B-E). Collectively, our data support delayed engulfment and clearance of dead hepatocytes (efferocytosis) at intermediate stages (2-5 days post injury), and eventual departure of macrophages at late stages (5-8 days post-injury) as responsible for mito-DAMPs “leakage” from persisting dead hepatocytes in BALBc. Such inefficient efferocytosis in BALBc mice explains persistence of macrophage infiltrate within necrotic areas in late recovery stages (Fig. S3B-D). We revised main text extensively to describe our results on recruitment versus phagocytic function in a more specific and unambiguous manner, and to include new data.

REVIEWERS' COMMENTS:

Reviewer #1 (Remarks to the Author):

I have no additional comments for the authors that have addressed all Reviewer concerns.

Reviewer #2 (Remarks to the Author):

Authors effectively addressed the comments from previous reviews by adding several new data. The overall study has been improved significantly. No further major concerns.

Minor comments:

1. l.173 "Fig.5B&C" is incorrect.
2. The title of Figure legend S4 should be fixed.

Point by point response to reviewer's comments:

Please find below our detailed responses to specific reviewers comments. Relevant minor changes were tracked the text of the revised manuscript.

Reviewer #1:

I have no additional comments for the authors that have addressed all Reviewer concerns.

A: We are glad that all reviewer's comments were addressed satisfactorily, and thank the expert reviewer for his time and effort to evaluate our manuscript.

Reviewer #2:

Authors effectively addressed the comments from previous reviews by adding several new data. The overall study has been improved significantly. No further major concerns.

Minor comments:

- 1. l.173 "Fig.5B&C" is incorrect.*
- 2. The title of Figure legend S4 should be fixed.*

A: We are glad that all reviewer's comments were addressed satisfactorily, and thank the expert reviewer for his time and effort to evaluate our manuscript. Minor comments were corrected as requested.